# Investigation on the Seismic Wave Propagation Characteristics Excited by Explosion Source in High-Steep Rock Slope Site Using Discrete Element Method

Danqing Song [1,2], Xuerui Quan [3], Mengxin Liu [4,*], Chun Liu [3,*], Weihua Liu [5], Xiaoyu Wang [4] and Dechao Han [5]

1   School of Civil Engineering and Transportation, South China University of Technology, Guangzhou 510640, China
2   State Key Laboratory of Subtropical Building Science, South China University of Technology, Guangzhou 510640, China
3   School of Earth Sciences and Engineering, Nanjing University, Nanjing 210023, China
4   School of Civil Engineering, Northeast Forestry University, Harbin 150040, China
5   SINOPEC Key Laboratory of Geophysics, Nanjing 211103, China
*   Correspondence: liumengxin@nefu.edu.cn (M.L.); chunliu@nju.edu.cn (C.L.)

**Abstract:** The influence of seismic waves induced by explosion sources on the dynamic response characteristics of rock slope sites is one of the most important problems affecting engineering construction. To investigate the wave propagation characteristics and attenuation law of seismic waves induced by explosive sources in rock sites from the perspective of time and frequency domains, the high-performance matrix discrete element method (MatDEM) is used to carry out numerical simulation tests on a granite rock medium site. The discrete element model of the high-steep rock slope is established by MatDEM, and the dynamic analysis of the rock medium site is conducted by loading blasting vibration load to generate seismic waves. The results show that the seismic waves in the rock site present characteristics of arc propagation attenuation. The maximum attenuation rate of the dynamic response is the fastest within 0.3 s and 25 m from the explosion source. The slope region can weaken the dynamic response of seismic waves generated by the explosion source. In particular, the high-frequency band (>20 Hz) has an obvious filtering effect. The dynamic response of the P-wave induced by the explosive source is greater than that of the S-wave in the bedrock and surface region. The dynamic amplification effect of the P-wave is greater than that of the S-wave in the slope region. The seismic waves in the slope region show an attenuation effect along the slope surface and have a typical elevation amplification effect inside the slope.

**Keywords:** seismic wave characteristics; attenuation law; explosion source; time and frequency domains; rock slope site

## 1. Introduction

The rock breaking method of blasting has the advantages of low cost, fast construction, and high efficiency, and has become the main method for the construction of basic engineering, such as slope and tunnel driving [1–4]. Seismic waves triggered by explosive blasting will induce slope instability, deformation, and destruction of rock-soil mass and other disasters, seriously threatening people's lives and property safety [5]. In addition, seismic exploration technology is one of the most important methods to explore oil resources, and more than 90% of oil reserves in China are explored using this technology [6]. At present, seismic exploration is faced with difficulties, such as difficulty and hidden storage, which will inevitably put forward higher requirements for exploration resolution [7]. Hence, the seismic effect of explosive blasting has become a key problem in land exploration and engineering construction. Explosion excitation seismic wave is one of the most active research topics in the field of engineering blasting.

At present, field blasting experiments, vibration tests, and numerical simulations are mainly used to analyze and investigate the propagation characteristics and attenuation law of blasting vibration [8]. In particular, numerical simulation has become one of the commonly used methods in the field of explosion impact [9]. The finite element method (FEM), finite difference method (FDM), and discrete element method (DEM) are commonly used in numerical simulations of explosion impact [10,11]. Some scholars have researched the propagation characteristics and attenuation law of blasting seismic waves by using FEM numerical simulation method [12,13]. FEM has great limitations for discontinuous media, infinite domain, large deformation, and stress concentration. Aiming at the large deformation of discontinuous media, some scholars began to use the FDM to investigate the dynamic response characteristics of rock-soil mass [14,15]. However, it is difficult to simulate the failure process of the rock-soil mass in the FDM because of its arbitrary division and boundary conditions. Some scholars began to use the DEM to study the dynamic response law of rock slopes, such as the PFC2D [16], UDEC [17], and so on. Moreover, MatDEM has been applied to geotechnical engineering by some scholars, in particular, the slope disaster process analysis has been discussed via MatDEM. Chen and Song used MatDEM to study the evolution process and the failure of mode of loess bedrock landslide [18]. Chen et al. used MatDEM to simulate the failure evolution process of reservoir bank landslide and to forecast the danger areas affected by reservoir bank landslide [19]. Li et al. Used MatDEM to study the deformation and failure process and evolution of fractured rock mass [20]. The results show that the DEM is mainly suitable for solving discontinuous media and large deformation problems, and can better simulate the dynamic response and failure process of complex rock-soil mass. Therefore, many achievements have been made in the numerical simulation of explosive source seismic waves in previous studies. However, due to the limitations of the previous DEM, such as particle number, computing speed, and computing efficiency, a more efficient and convenient discrete element numerical simulation method is urgently needed.

There are many factors affecting the propagation characteristics of explosive source seismic waves [21]. Blasting vibrations involve not only the interaction between dynamite and rock mass, but also the propagation of the seismic wave in the stratum [22]. The influence of the explosion source on the propagation law of excitation seismic waves is relatively complex, and the process of excitation of a seismic wave by the explosive source is shown in Figure 1 [23]. There is a close relationship between rock-soil medium and seismic wave propagation of explosive source. The seismic wavelet generated by dynamite in practice is a very complex problem, and the seismic wavelet is related to the property of strata rock, which is itself a complex body [24]. Many studies have shown that the characteristics of rock mass have become one of the main factors affecting the propagation characteristics of explosive source seismic waves, including rock type, rock weathering, groundwater level, topography, geomorphology, and other factors [25]. Some scholars have researched the influence of landforms on the propagation law of blasting vibration waves and the research results mainly focus on the blasting vibration and attenuation law under flat terrain, the attenuation effect of concave landforms, the amplification effect of convex landforms, etc. [26–28]. At present, the propagation characteristics of explosion source excited wavelets in rock slope sites need to be further explored. Given the above-mentioned analysis, the law of explosive source seismic wave propagation in rock medium sites needs to be further explored. It is necessary to further reveal the propagation characteristics, amplification effect, and attenuation law of explosive source seismic waves in rock medium sites. At the same time, there is a lack of efficient discrete element numerical simulation methods and algorithms to realize the numerical simulation of explosive source excitation seismic wave.

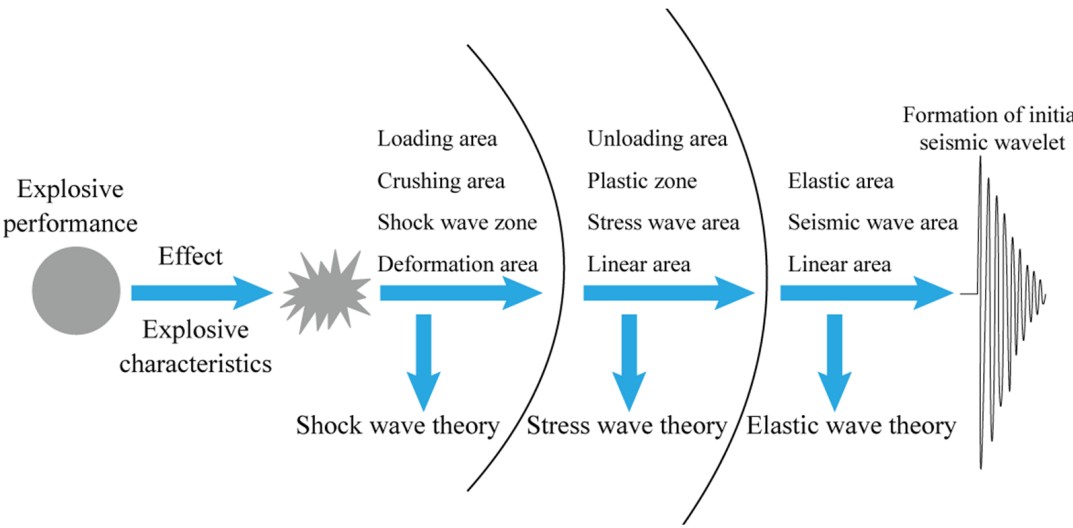

**Figure 1.** Explosive sources excite seismic wave processes Adapted with permission from Ref. [23]. 2015, Mou J.

In this work, fast GPU Matrix computing of the Discrete Element Method (MatDEM) was used to conduct numerical simulation tests. The discrete element numerical model of high-steep rock slope is established by using MatDEM, and the discrete element dynamic analysis of the rock medium site is conducted by loading blasting vibration load and generating a seismic wavelet. By analyzing the distribution of velocity and displacement fields, and the Fourier spectrum characteristics of explosive source wavelet, the time-frequency analysis method is used to explore the wave propagation characteristics of explosive source wavelet in rock medium field from the perspective of time and frequency domains. By analyzing the spatial variation law of key parameter peak value of seismic waves in time and frequency domains, their waveform characteristics in rock slope sites are studied. The wave propagation characteristics, field amplification effect, and attenuation law of explosive source wavelet are explored as well. This work can deepen the understanding of the propagation characteristics and disaster-causing mechanism of explosive source seismic waves in rock medium under complex conditions, which has important scientific significance and application value. This work takes the typical granite lithologic medium site as the research object. The research mainly focuses on the propagation characteristics of seismic source wavelets in the rock slope site under the blasting effect of explosives.

## 2. Methodology

### 2.1. Fast GPU Matrix Computing of the Discrete Element Method (MatDEM)

MatDEM technology [29,30] is similar to particle flow code (PFC), which is a DEM software developed by Cundall and Strack [31] (1979) to simulate particle assemblage. The program uses a series of discrete units that can be closely grouped to simulate a rock-soil mass. These elements can be bonded together with linearly elastic springs, normal springs and shear springs, and the bonds between them carry loads. In MatDEM, viscous damping is used to dissipate kinetic energy. The MatDEM model is similar to the shell model, which can monitor the physical and mechanical behavior of rock and soil mass during simulation. MatDEM is based on the matrix discrete element method, the calculation is fast, and it is the foundation and core of the whole software. MatDEM adopts the innovative GPU matrix calculation method and 3D contact algorithm to realize the calculation of 3D cell motion 14 million times per second (40 million two-dimensional), and the number of calculation cells and calculation speed reaches 30 times that of foreign commercial software (1.5 million units). This work develops a high-performance matrix discrete element numerical method for calculating the propagation characteristics of seismic waves in rock slope sites under blasting excitation. The high-performance discrete element numerical simulation of millions

of particles is achieved via the innovative matrix discrete element calculation method. The extension of discrete element analysis from the sample scale to the engineering slope scale can automatically train materials, conduct a numerical simulation of blasting impact, and realize the quantitative analysis of complex rock and soil problems. From the material property, velocity field, displacement field, and energy distribution, the wave propagation characteristics and attenuation law of rock slope under blasting are effectively and truly revealed. The specific modeling process is shown in Figure 2.

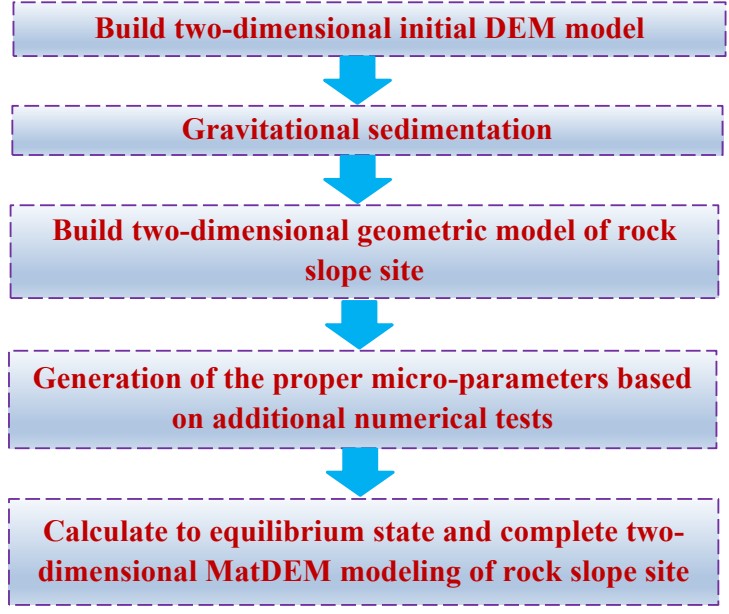

**Figure 2.** Flowchart of MatDEM model construction of two-dimensional rock slope site.

The discrete element method uses randomly stacked particles to establish a discrete element model, which assumes that the rock-soil mass is composed of a large number of elastic particle units stacked. There are normal force, tangential force, and surface friction between particles [28,29]. The normal spring force ($F_n$) is expressed as:

$$F_n = \begin{cases} K_n X_n, X_n < K_n; \text{ The connections between particles are intact.} \\ K_n X_n, X_n < 0; \text{ The connections between particles are intact.} \\ 0, X_n > 0; \text{ The connections between particles are intact.} \end{cases} \quad (1)$$

where $K_n$ is the normal stiffness between elements; $X_n$ is the normal displacement between elements (Figure 3); and $X_b$ is the inter-element fracture displacement.

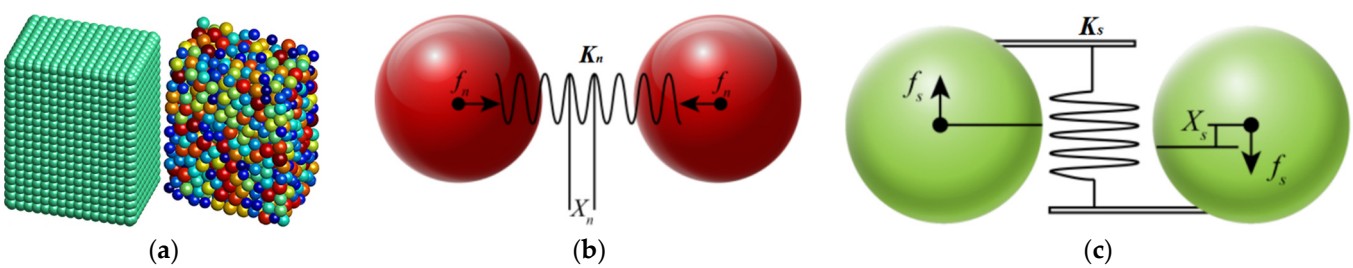

**Figure 3.** Schematic diagram of linear elastic model: (**a**) discrete element accumulation body; (**b**) inter-particle normal force; (**c**) inter-particle shear force. Adapted with permission from Ref. [29]. 2019, Liu C, et al.)

As shown in Formula (1), when elements are connected, there is a normal interaction force between elements if they are pulled and pressed by external forces. If the element is

subjected to tension, the external force increases, the distance increases, and there is fracture displacement $X_n$. After the connection is broken, the elements no longer interact with each other. When compressed, repulsive forces are generated between elements [29,30]. The tangential spring force ($F_s$) is also consistent with normal spring force and can be expressed as:

$$F_n = K_s X_s \tag{2}$$

where the $K_s$ is the tangential stiffness; and the $X_s$ is the tangential displacement.

In the tangential direction, there exists the maximum shear force $F_{smax}$, which can be calculated according to the Mohr-Coulomb criterion when the cemented connection between elements is intact:

$$F_{smax} = F_{s0} - \mu_p \cdot F_n \tag{3}$$

where the $F_{s0}$ is shear resistance between elements; $\mu_p$ is the friction coefficient between elements; and $F_n$ is the normal force. When the tangential external force on elements exceeds $F_{smax}$, the tangential connection between elements breaks, and the maximum shear force becomes $F_{smax}{}'$:

$$F_{smax}{}' = -\mu F_n \tag{4}$$

The numerical calculation of the discrete element method is based on the time-step iterative algorithm. In a very small time-step, the force between each element and its adjacent elements is calculated, and the resultant force of the element, acceleration, velocity, displacement, and new stress state are obtained. After the calculation of the current time-step is completed, another time-step is advanced to realize the iteration of the discrete element method. Based on the above principles, this work developed a vibration boundary condition input function and vibration monitoring point recording function based on discrete element software MatDEM through the secondary development function of the software and according to the requirements of blasting dynamic action analysis [32]. Through this work, various vibration boundary conditions can be applied, and the information data of monitoring points can be recorded and used to analyze the amplification effect of ground motion.

### 2.2. Discrete Element Numerical Model of High-Steep Rock Slope

The establishment of the rock slope site DEM model mainly includes two steps: random stacking and cutting modeling. First, random units are generated, and gravity deposition and compaction are conducted to simulate the diagenetic process of rocks in nature. The established model is 1500 m in length and 600 m in height. The particle unit has a radius of 1.2 m and contains a total of 589,247 units. The elements outside the slope and bedrock area on the right of the model were deleted, and the model stress balance was calculated. The stratum and slope materials refer to the macroscopic mechanical properties of a natural granite material (Table 1), and the corresponding microscopic mechanical parameters are obtained through the conversion formula (Table 2). The transformation formula of macro and micromechanical properties of the discrete element model is as follows:

$$K_n = \frac{\sqrt{2}Ed}{4(1-2v)} \tag{5}$$

$$K_s = \frac{\sqrt{2}(1-5v)Ed}{4(1+v)(1-2v)} \tag{6}$$

$$X_b = \frac{3K_n + K_s}{6\sqrt{2}K_n(K_n + K_s)} \cdot T_u \cdot d^2 \tag{7}$$

$$Fs_0 = \frac{1-\sqrt{2}\mu_p}{6} \cdot C_u \cdot d^2 \tag{8}$$

$$\mu_p = \frac{-2\sqrt{2}+\sqrt{2}I}{2+2I}, \ I = \left[\left(1+\mu_i^2\right)^{1/2} + \mu_i\right]^2 \tag{9}$$

**Table 1.** Macro-parameters of ggranite under natural conditions.

| Type | Density (kg/m$^3$) | Young's Modulus $E$ (MPa) | Poisson's Ratio $v$ | Uniaxial Tensile Strength $T_u$ (kPa) | Uniaxial Compression Strength $C_u$ (kPa) | Coefficient of Internal Friction |
|---|---|---|---|---|---|---|
| 2800 | $3.5 \times 10^4$ | 0.26 | $1.0 \times 10^4$ | $2.0 \times 10^5$ | 1.0 | 2800 |

**Table 2.** Average element mechanical parameters of ggranite in the discrete element model.

| Type | Tangential Stiffness $K_s$/(kN/m) | Normal Stiffness $K_n$/(kN/m) | Fracture Displacement $X_b$/m | Shear Resistance $F_{s0}$/GN | Coefficient of Friction $\mu_p$ | Cell Radius/m | Total Number of Units |
|---|---|---|---|---|---|---|---|
| Granite | $3.39 \times 10^7$ | $1.96 \times 10^8$ | $3.89 \times 10^{-4}$ | $3.81 \times 10^{-1}$ | 0.40 | 1 | 204,667 |

For the linear elastic model, the normal stiffness ($K_n$), tangential stiffness ($K_s$), fracture displacement ($X_b$), initial shear force ($F_{s0}$), and friction coefficient ($\mu_p$) can be represented by five macroscopic mechanical properties of materials. This means that Young's modulus ($E$), Poisson's ratio ($v$), compressive strength ($C_u$), tensile strength ($T_u$), and coefficient of internal friction ($\mu_i$) are calculated by the above formula. In the above formula, d is the diameter of the element.

To investigate the influence of the rock slope site on the propagation characteristics and attenuation rule of the blasting source, slope site models with a slope angle of 60° and slope height of 200 m were established, respectively, as shown in Figure 4. The information recording module of explosion-induced seismic waves built into MatDEM is used to set 67 monitoring points in the valley model (Figure 4c) to record the acceleration time history data of the seismic waves triggered by blasting sources at different positions in the slope and bedrock region. The layout scheme of measuring points is based on the following two considerations: First, the purpose of laying measuring points on the horizontal and vertical axis of the explosion source is to study the wave propagation characteristics and attenuation law as the distance from the explosion source increases. Second, to study the influence of the sloping site on the propagation characteristics and attenuation law of the seismic waves triggered by the explosion source, the measuring points are set up on the surface, inside, and around the slope crest. Third, 67 measuring points are laid according to different regions to extract the PGA and PGA of seismic waves in different regional positions, to facilitate the drawing of seismic wave PGA by Surfer software based on interpolation method. The attenuation law of seismic waves is studied by analyzing the distribution characteristics of PGA. In the numerical simulation, the blasting focal location is measuring point 27 (MP-27), located 100 m away from the slope toe. Blasting load dynamite of 2172.67 kg is applied to MP-27 (explosion source) for blasting to excite seismic waves. Based on the above settings, MatDEM is used for numerical simulation to analyze the propagation characteristics and attenuation law of seismic waves in the sloping site.

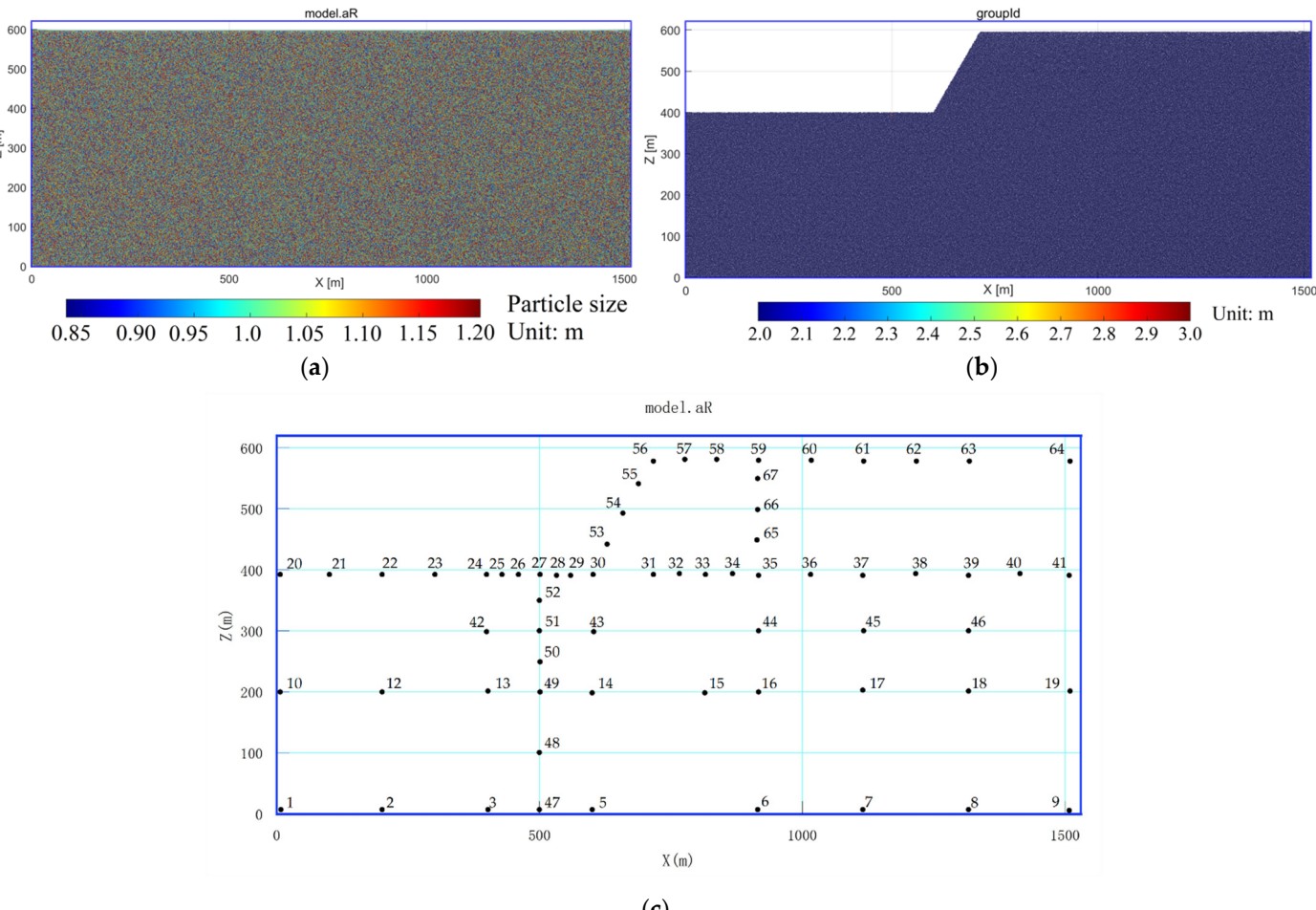

**Figure 4.** DEM model: (**a**) Model box; (**b**) DEM model (MatDEM) for the slope; (**c**) Layout of measuring points.

### 2.3. Time-Frequency Analysis Method

In this work, the time domain research content mainly includes seismic wave propagation characteristics, dynamic amplification effect, and so on. The main time-domain parameters include the distribution of velocity/displacement wave fields, the velocity/acceleration time history curve, etc. Frequency domain analysis includes the Fourier spectrum characteristics and natural frequency analysis. The analysis results of time and frequency domains verify and complement each other. The propagation characteristics and attenuation law of seismic waves induced by explosion source in the rock slope site are studied from the perspective of multiple domains. In addition, the Fast Fourier transform (FFT) is a combination of seismic signals into multiple harmonic signals, which is widely used in signal analysis and processing [32]. Frequency domain analysis mainly includes Fourier spectrum analysis and power spectrum analysis. Based on frequency domain signal analysis, the spectrum response of rock and soil mass and spectrum characteristics of seismic waves can be obtained. Fourier transform has the advantages of good frequency positioning and clear identification of different frequency components of the signal. FFT can quickly identify the main components of the signal, filter, and so on, and become a common means of processing seismic signals. The mathematical expression of FFT is as follows [33,34]:

$$F(a) = \int_{-\infty}^{+\infty} x(t)e^{-j2\pi at}dt \tag{10}$$

where $a(t)$ is the acceleration history in the time domain, and $F(a)$ is the Fourier transform of the acceleration history $a(t)$.

## 3. Results

### 3.1. Wave Propagation Characteristics of Seismic Waves Triggered by Explosion Source in the Time Domain

#### 3.1.1. Propagation Characteristics of the Seismic Wave Field

To study the propagation characteristics of seismic waves induced by explosion source in rock slope site, the velocity and displacement wave field distribution in the discrete element model is analyzed as an example. Figures 5 and 6 show the distribution of velocity and displacement wave field under blasting. Figure 5 shows that under the blasting action, the velocity wave is centered at the explosion point and transmitted in an arc from near to far along the bedrock area. The initial seismic wave was excited by the explosion, which excited a velocity wave with a peak value of approximately 30 m/s at the explosion source (Figure 5a), and the amplitude of the outermost excited wave was about 15 m/s. As the explosive source time continued, the amplitude of the excited seismic wave gradually decreased during the propagation process, and the amplitude of the outermost excited wave decreased to about 9 m/s (Figure 5b). With the continuation of blasting, the seismic wave induced by blasting gradually attenuates, and the amplitude of the outermost velocity wave is about 3 m/s (Figure 5c). When the velocity wave is transmitted to the bottom of the model and the boundary on both sides, the amplitude of the velocity wave shows further attenuation phenomenon (Figure 5d). Therefore, according to the distribution characteristics of the velocity field in the model, it can be seen that the seismic wave exhibits the characteristics of arc-shaped transmission, and the amplitude of the velocity wave continues with the time of the explosion. With the increase in distance from the explosion source, the velocity wave appears a gradual attenuation phenomenon.

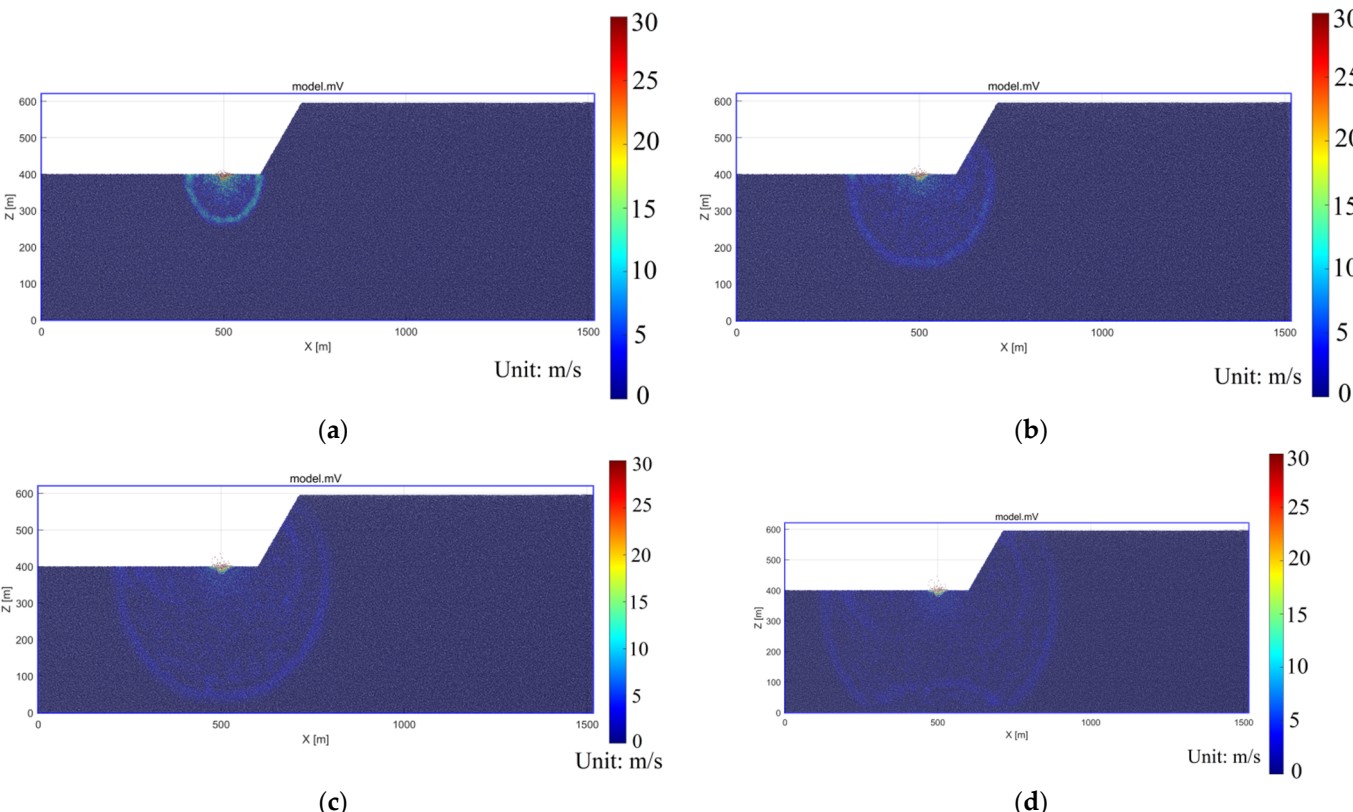

**Figure 5.** Distribution characteristics of the wavelet velocity field of explosion source in the DEM model. (**a**) t = 0.038 s; (**b**) t = 0.076 s; (**c**) t = 0.114 s; (**d**) t = 0.152 s.

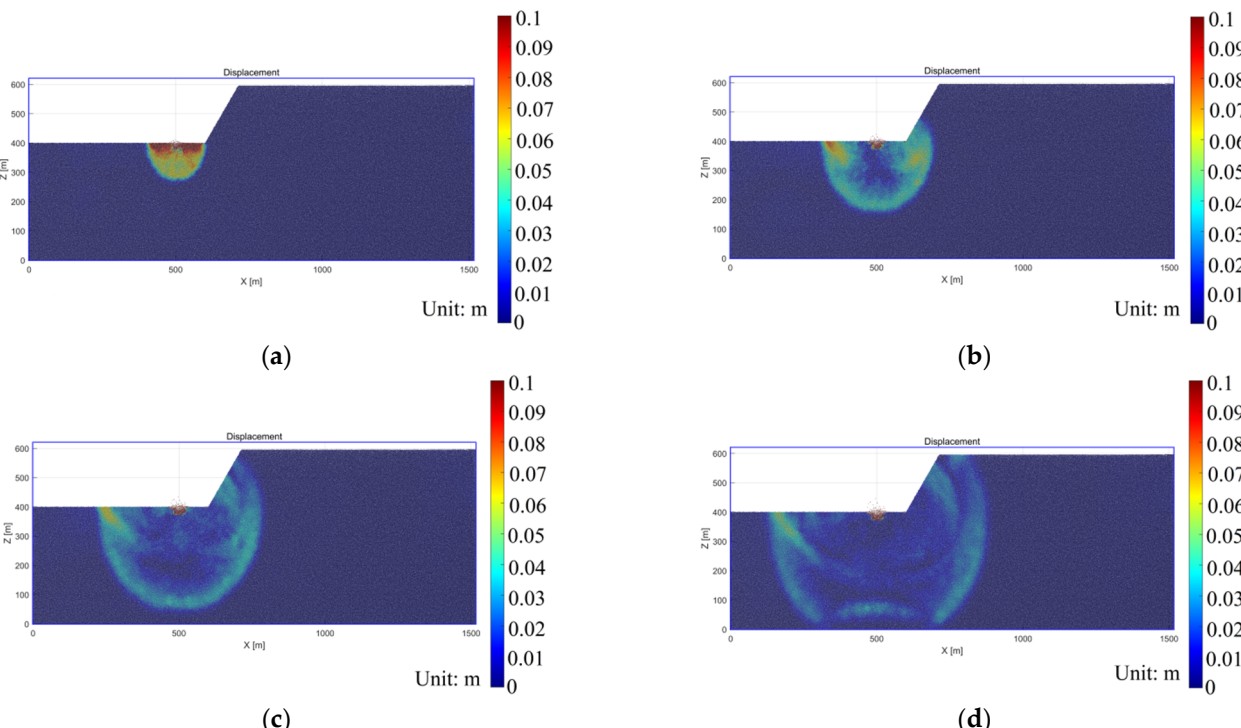

**Figure 6.** Distribution characteristics of wavelet displacement field of explosion source. (**a**) t = 0.038 s; (**b**) t = 0.076 s; (**c**) t = 0.114 s; (**d**) t = 0.152 s.

In addition, to further analyze the wave propagation characteristics of seismic waves triggered by the explosion source, the distribution of the displacement wave field is shown in Figure 6. Figure 6a shows that under the initial action of blasting, the displacement amplitude of the surface area around the explosion source appears a rapid amplification phenomenon. Figure 6b shows that with the duration of the explosion source time, the displacement wave propagates in a circular arc, and the amplitude of the displacement wave gradually decays. In particular, the displacement amplitude in the outermost area near the surface of the wave field is larger than that in the bedrock area. This indicates that the ground motion energy attenuation of the ground surface is weak compared with that of the bedrock area. After the displacement wave is transferred to a certain distance, the seismic wave reflects and scatters to a certain extent due to the topography fluctuation and cracks between particles in the slope area. The superposition or reduction effect between the reflected wave and the seismic wave generated by the explosion source causes the disorder of the wave field in the model, and the wave field is no longer transmitted with the circular arc characteristics. This is similar to the transmission characteristics of the velocity wave field.

### 3.1.2. Waveform Characteristics of Waves Triggered by Explosion Source

A blasting source excitation seismic wave is a kind of pulse signal with limited energy, fast attenuation speed, and concentrated amplitude energy. To investigate the waveform characteristics of seismic waves, the horizontal velocity and acceleration waves of the discrete element model are selected for analysis. In Figure 7a, the explosion source is measuring point 27 (MP-27). As the distance from the source increases, the velocity wave on the right side of MP-27 gradually attenuates, and the velocity amplitude attenuates from 26 m/s to 3 m/s within 0–0.4 s. After the focal vibration lasts for 0.03 s, the excited seismic waveform gradually becomes stable. Figure 7b,c show that the velocity amplitude on the left and lower side of the explosion source (MP-27) is relatively small, which attenuates from 17 m/s to 3 m/s within 0–0.08 s, and the seismic waveform gradually becomes stable with the continuous explosion shaking. By comparing Figure 7a with Figure 7b,c, the measuring

point in Figure 7a is close to the slope area, the amplitude of the seismic wave induced by the explosion source is affected by the slope area, and the amplitude of the seismic wave induced by the slope topographic relief is larger. In addition, the acceleration-time history of seismic waves is shown in Figure 8. The peak ground acceleration (PGA) of MP-27 is about 30.5 g, indicating that the acceleration at the explosion source is much larger than that at other locations. The seismic wave acceleration amplitude on the right side of the MP-27 rapidly decays from 10.5 g to 2.5 g within 0–0.03 s (Figure 8a). The maximum PGA of the measuring point on the left side of the explosion source is about 4.8 g (Figure 8b), and that of the measuring point on the lower side is about 1.3 g (Figure 8c). It can be seen that the PGA has the following characteristics: The PGA on the right side of the explosion source is the largest, but decays rapidly; in particular, the PGA shows a rapid attenuation from MP-28 to MP-29. The PGA of the seismic wave on the left and lower side of the explosion source is smaller than that on the right side of the explosion source, and the PGA below the explosion source is the smallest. That is, $PGA_{Right\ of\ explosion\ source} > PGA_{Left\ of\ explosion\ source} > PGA_{Below\ the\ explosion\ source}$, which indicates that the sloping topography has a certain amplification effect on the waves generated by the explosion source, and the attenuation rate of the seismic waves propagating near the surface is less than that in the bedrock.

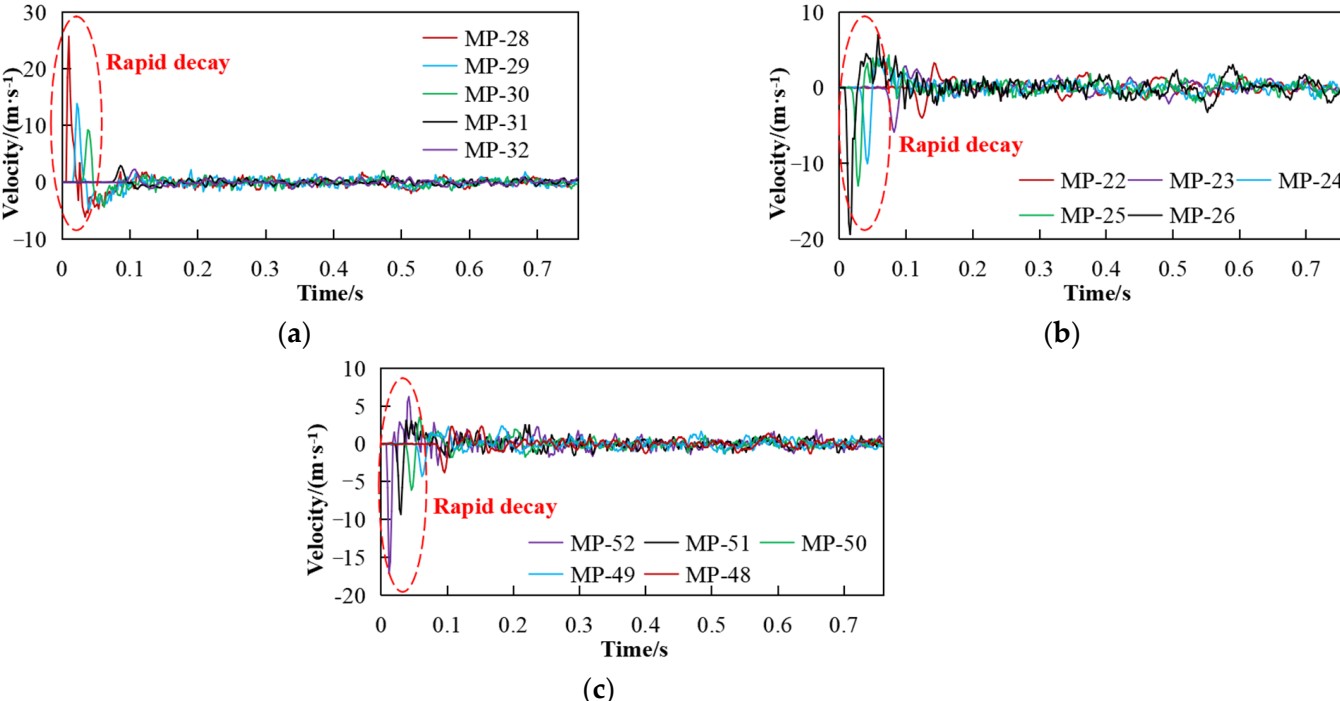

**Figure 7.** Propagation characteristics of horizontal velocity wavelet excitation from the explosion source: (**a**) Right side of the explosion source; (**b**) Left of the source of the explosion; (**c**) Below the source of the explosion.

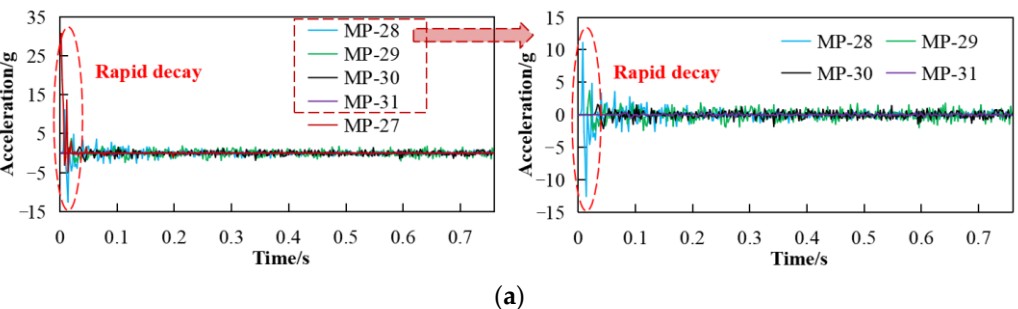

**Figure 8.** *Cont.*

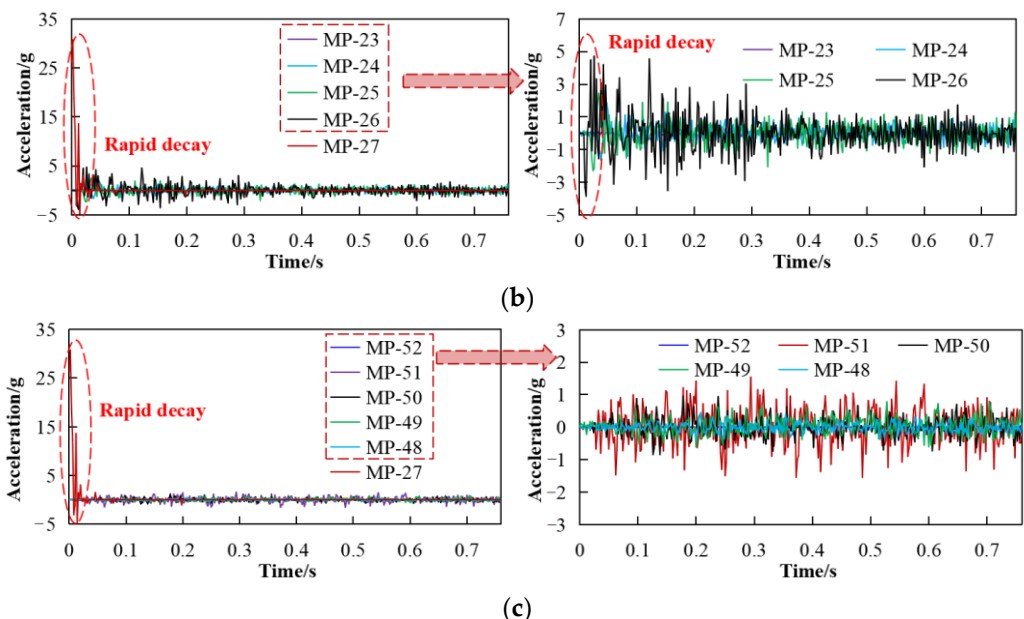

**Figure 8.** Propagation characteristics of horizontal acceleration wavelet excitation from the explosion source: (**a**) Right side of the explosion source; (**b**) Left of the source of the explosion; (**c**) Below the source of the explosion.

### 3.1.3. Dynamic Response Characteristics of S- and P-Waves of Seismic Wave Triggered by Explosion Source

The slope area is the focus area of research in this work. By analyzing the velocity and acceleration response characteristics of the slope surface and interior, the dynamic response characteristics and attenuation law of the seismic wave generated by the explosion source in the slope can be clarified. The seismic waves stimulated by the explosion source have directional properties, including transverse waves (S-wave), and longitudinal waves (P-wave). To study the waveform characteristics and attenuation rules of S-waves and P-waves generated by the explosion source in the slope area, the velocity and acceleration time-histories of the two kinds of waveforms were compared, as shown in Figures 9 and 10. Figure 9a shows that the PGV (Peak ground velocity) of the S-wave at the slope toe (MP-30) is about 9.2 m/s, which is larger than other measuring points on the slope surface, and rapidly attenuates along the slope surface with the increase in elevation. Figure 9b shows that the PGV of P-wave at the slope toe (MP-30) is about 4.8 m/s, larger than other measuring points on the slope surface, and slowly attenuates along the slope surface. Figure 10a shows that the PGA of the S-wave of MP-30 is about 2.4 g, which is significantly larger than other measuring points on the slope surface, and it decays rapidly along the slope surface. In Figure 10b, the PGA of the P-wave at the MP-30 is about 1.27 g, which is larger than other measuring points on the slope surface, and slowly decays along the slope surface. According to Figures 9 and 10, the attenuation law of S- and P-waves at the slope surface is similar to that of S- and P-waves inside the slope area, and the PGV and PGA of S-wave are greater than that of P-wave overall.

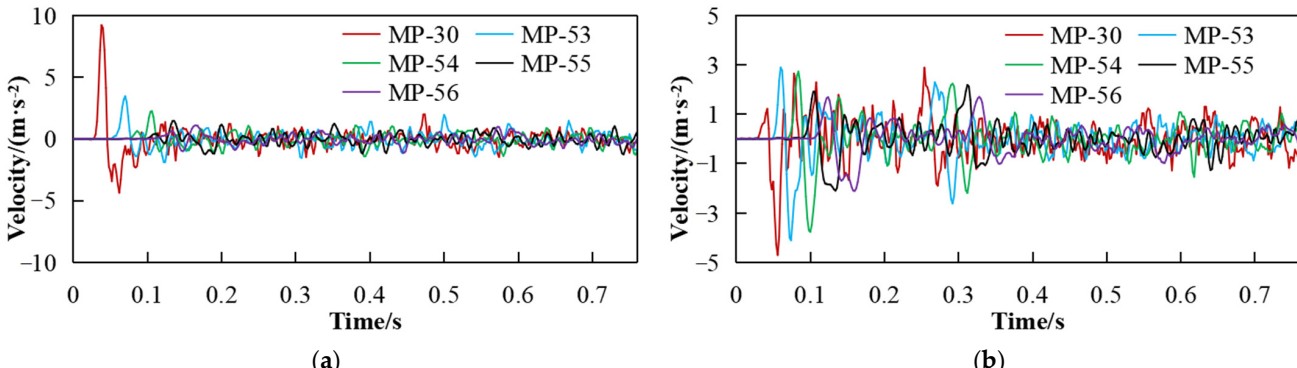

**Figure 9.** Velocity waveform of explosive source excited wavelet at the slope surface: (**a**) horizontal time histories; (**b**) vertical time histories.

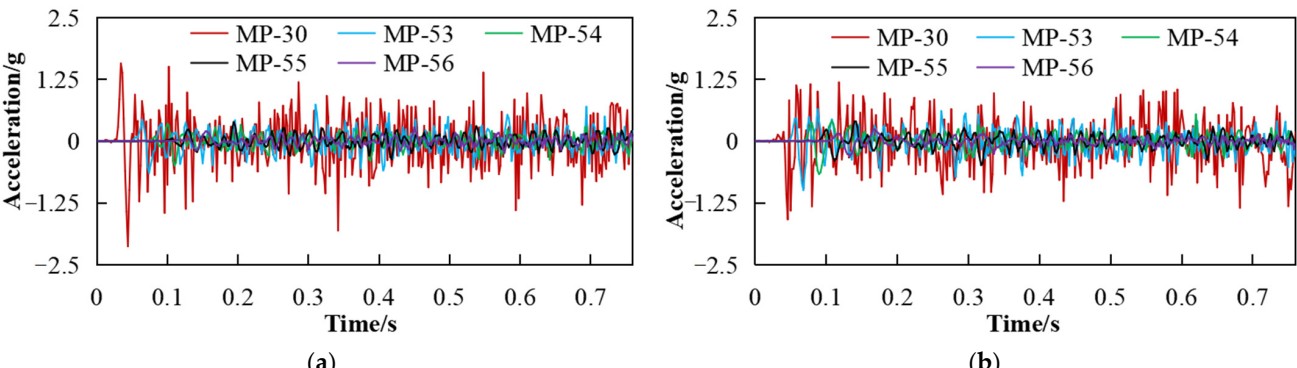

**Figure 10.** Acceleration waveform of explosive source excited wavelet at the slope surface: (**a**) horizontal time histories; (**b**) vertical time histories.

In addition, to further explore the dynamic response characteristics and attenuation rules of S- and P-waves, the $PGVx/PGVz$ and $PGAx/PGAz$ in different regions of the model are shown in Figure 11. $PGVx$ and $PGAx$ are the PGV and PGA of S-wave, and $PGVz$ and $PGAz$ are PGV and PGA of P-wave, respectively. According to Figure 11a, $PGVx/PGVz$ and $PGAx/PGAz$ in the slope region are less than 1.0. This indicates that the PGV and PGA of the S-wave in the slope region are smaller than that of the P-wave. Thus, the dynamic amplification effect of the S-wave is larger than that of the P-wave. Figure 11b–d shows that $PGVx/PGVz$ and $PGAx/PGAz$ along the horizontal and vertical axis of the explosion source are generally greater than 1.0. This indicates that the dynamic response of the P-wave is greater than that of the S-wave in the bedrock area and the surface area.

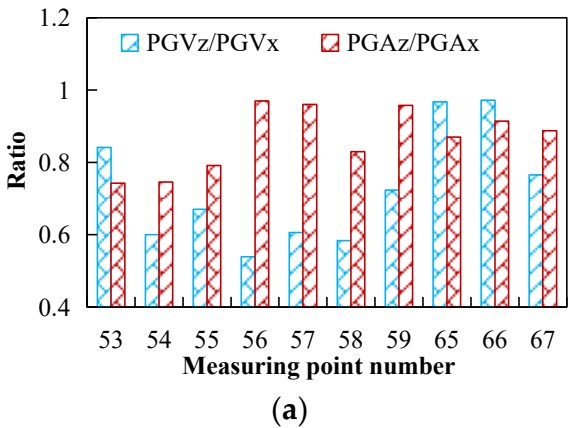

(**a**)

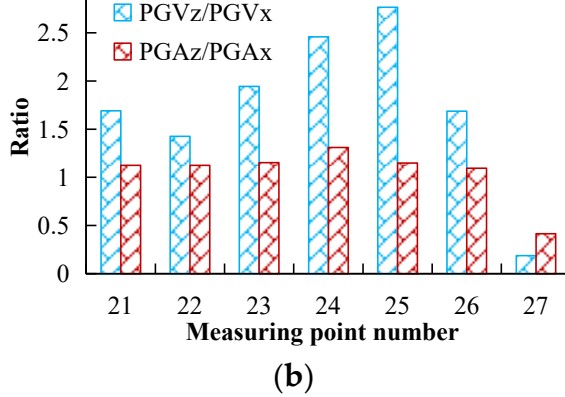

(**b**)

**Figure 11.** *Cont.*

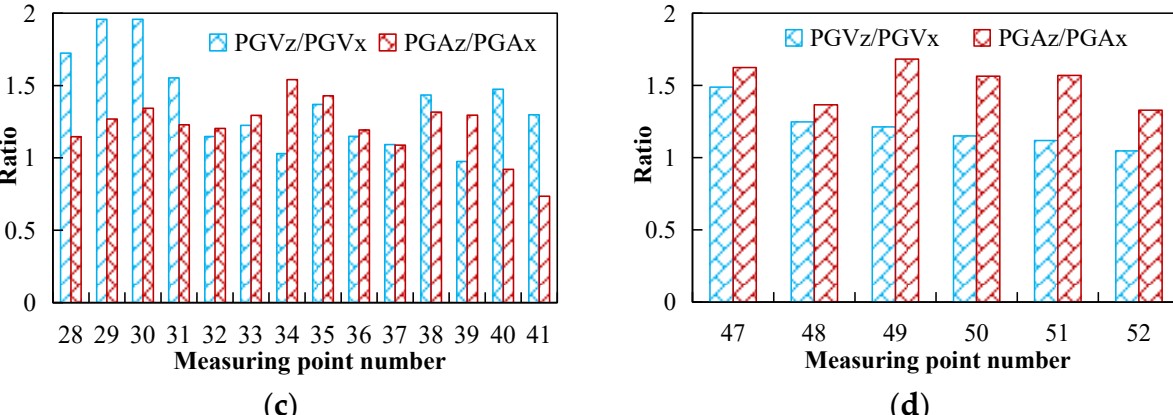

**Figure 11.** The ratio of horizontal peak to vertical peak in the DEM model: (**a**) Slope area; (**b**) Left side of the explosion source; (**c**) Right side of the explosion source; (**d**) Below the explosion source.

3.1.4. Attenuation Law of Seismic Wave Triggered by Explosion Source

The above research shows that the dynamic response characteristics and attenuation law of the S-wave and P-wave generated by the explosion source are different in the slope area and the bedrock area. To further study the attenuation law of seismic waves in the slope and bedrock area, the PGA and PGV of the representative measuring points of slope and bedrock are shown in Figures 12–14. Figure 12 shows that the PGA and PGV of the S-wave are significantly greater than those of the P-wave. The PGA and PGV of the slope surface decrease gradually along the increase in elevation and the phenomenon of seismic wave attenuation appears. However, in the slope area, PGA and PGV gradually grow larger along with the increase in elevation and the dynamic amplifying effect of seismic waves appears. This indicates that the seismic wave generated by the explosion source has an obvious elevation amplification effect in the inner region of the slope. Figures 13 and 14 show that, along the horizontal and vertical axis of the explosion source, the PGA and PGV gradually decrease with the increase in the distance from the source, showing an obvious attenuation effect. In particular, near the source, the PGV and PGA attenuation of the interior seismic wave is in the range of 1/8 to 1/5 of 25 m along the horizontal and vertical axes. In the bedrock area where the transverse and longitudinal wheelbase is more than 100 m from the explosion source, the PGV and PGA gradually decay and tend to stabilize. This indicates that the seismic wave generated by the explosion source has a limited range of influence and has the greatest influence on the dynamic response within the range of 25 m from the explosion source, while the influence is small in the area beyond 100 m from the explosion source. Moreover, to further highlight the attenuation characteristics of seismic waves stimulated by the blasting source, the distribution characteristics of PGA and PGV of seismic waves are shown in Figure 15. Figure 15 shows that the PGA and PGV generated by the explosion source are larger than those in other areas within a certain distance near the source. Overall, PGA and PGV decreased with the increase in distance from the source. In other words, the seismic wave generated by the explosion source has an obvious attenuation effect.

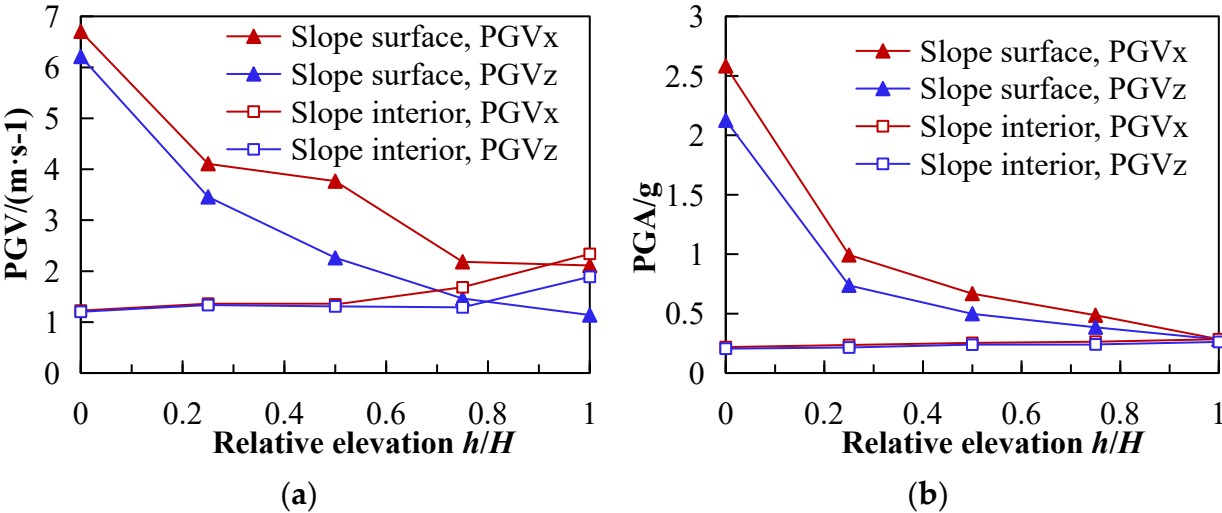

**Figure 12.** Dynamic amplifying effect in the slope area: (**a**) PGV; (**b**) PGA.

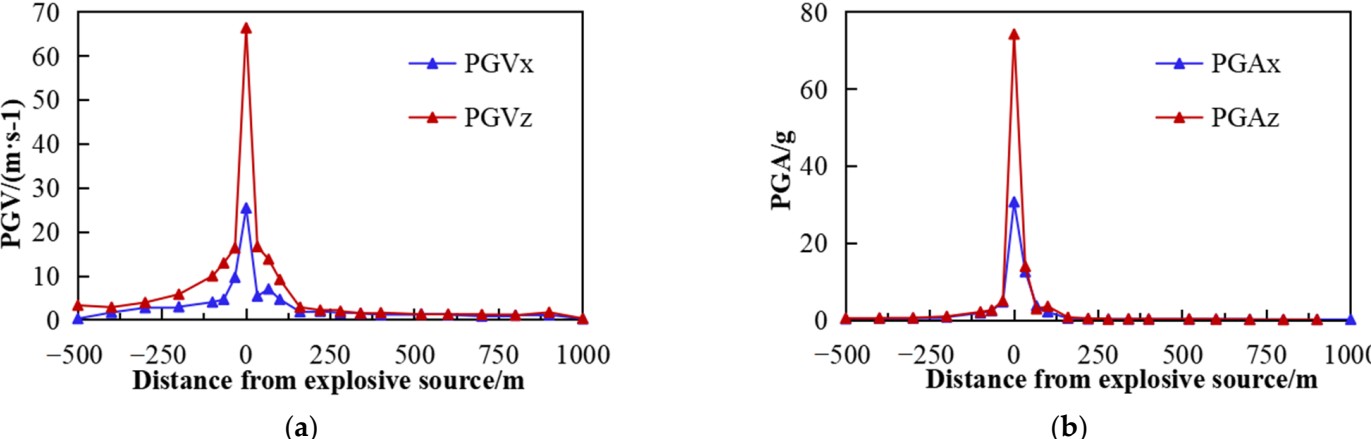

**Figure 13.** Characteristics of dynamic amplifying effect in horizontal axial direction of explosion source: (**a**) PGV; (**b**) PGA.

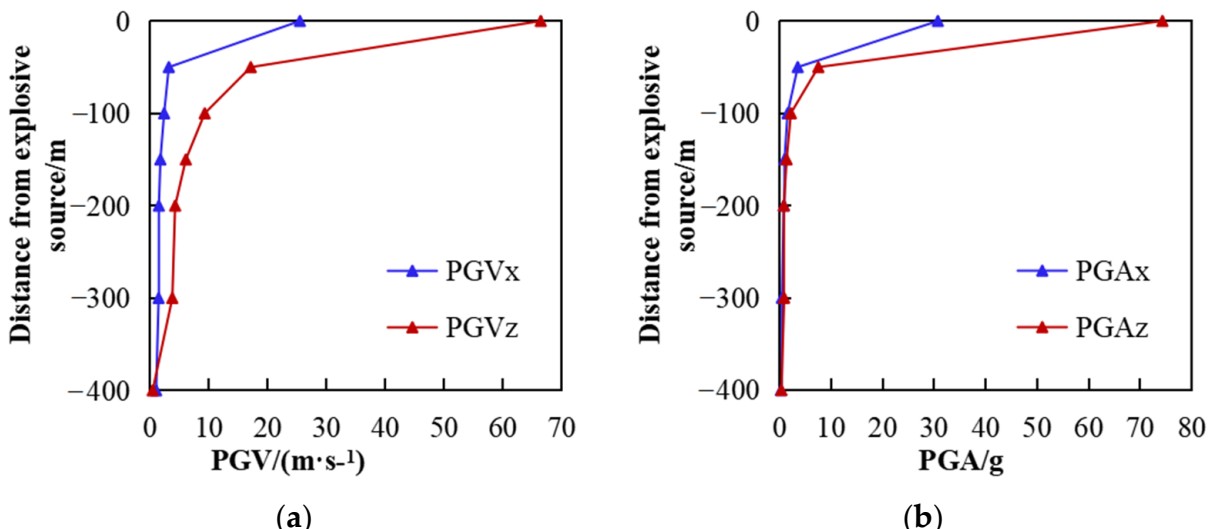

**Figure 14.** Characteristics of dynamic amplifying effect in the vertical axial direction of explosion source: (**a**) PGV; (**b**) PGA.

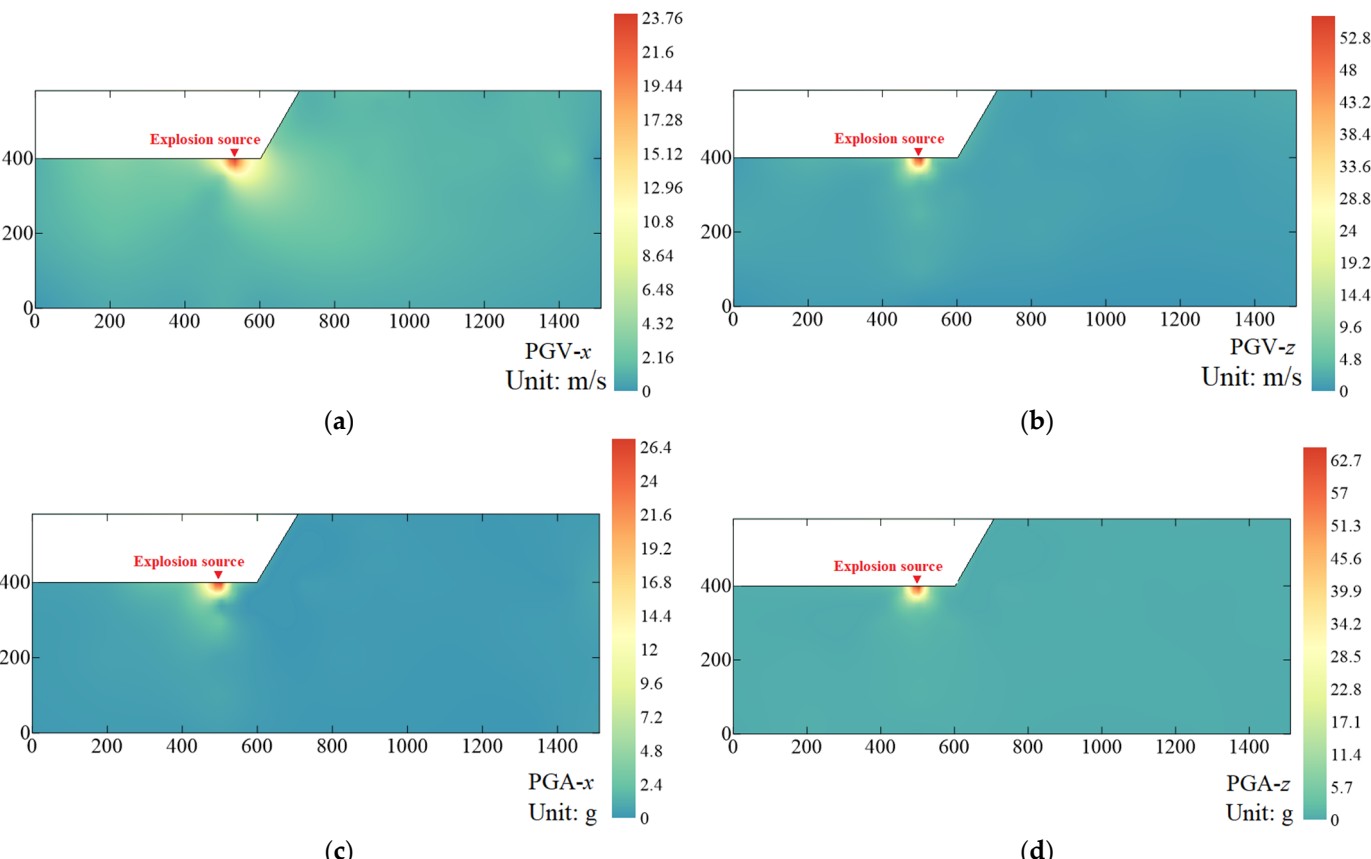

**Figure 15.** Wavelet peak value distribution of explosion source: (**a**) PGV*x*; (**b**) PGV*z*; (**c**) PGA*x*; (**d**) PGA*z*.

*3.2. Wave Propagation Characteristics of Waves Stimulated by Explosion Source in the Frequency Domain*

3.2.1. Fourier Spectrum Characteristics Analysis

The frequency components of seismic waves induced by blasting are complex, and it is difficult to fully reveal the wave propagation and dynamic response characteristics of waves in the rock slope site in the time domain. To explore the attenuation law of seismic waves induced by an explosion source in the slope from the frequency domain, the Fourier spectrum of the typical measuring points in the model is obtained by performing FFT on the acceleration-time history of waves. Taking the explosion source (MP-27) as the center, Fourier spectra of S-wave acceleration-time histories of the transverse/longitudinal axis of the explosion source and slope surface are shown in Figure 16. Figure 16 shows that the Fourier spectrum of measuring points around the explosion source and on the slope surface is rich in frequency components, and the identification of superior frequency is poor. In particular, the spectral characteristics of the bedrock area under the explosion source are more complex. This indicates that the bedrock region has no obvious filtering effect on the frequency components of the seismic waves at the explosion source.

In addition, to further explore the influence of slope on the frequency components of seismic waves, the acceleration time history of P-wave at the slope surface, inside the slope, and at the slope crest is taken as an example. The corresponding Fourier spectra are shown in Figure 17. Figure 17 shows that the Fourier spectrum features inside the slope are different from that of the slope surface, and the Fourier spectrum features of the sloping interior and slope crest have obvious superior frequency segments. The amplitude of the Fourier spectrum larger than 20 Hz on the frequency axis showed a large decrease phenomenon overall as the amplitude is less than 0.01. In addition, by comparing the Fourier spectrum amplitudes of the slope surface, slope interior, and on slope crest, it can

be found that the Fourier amplitudes of the sloping interior and slope crest are smaller than those of the slope surface. This indicates that the slope region has a weakening effect on the dynamic response of seismic waves, especially for the high frequency (>20 Hz) seismic wave, which has an obvious filtering effect, greatly reducing the amplitude of the high-frequency spectrum.

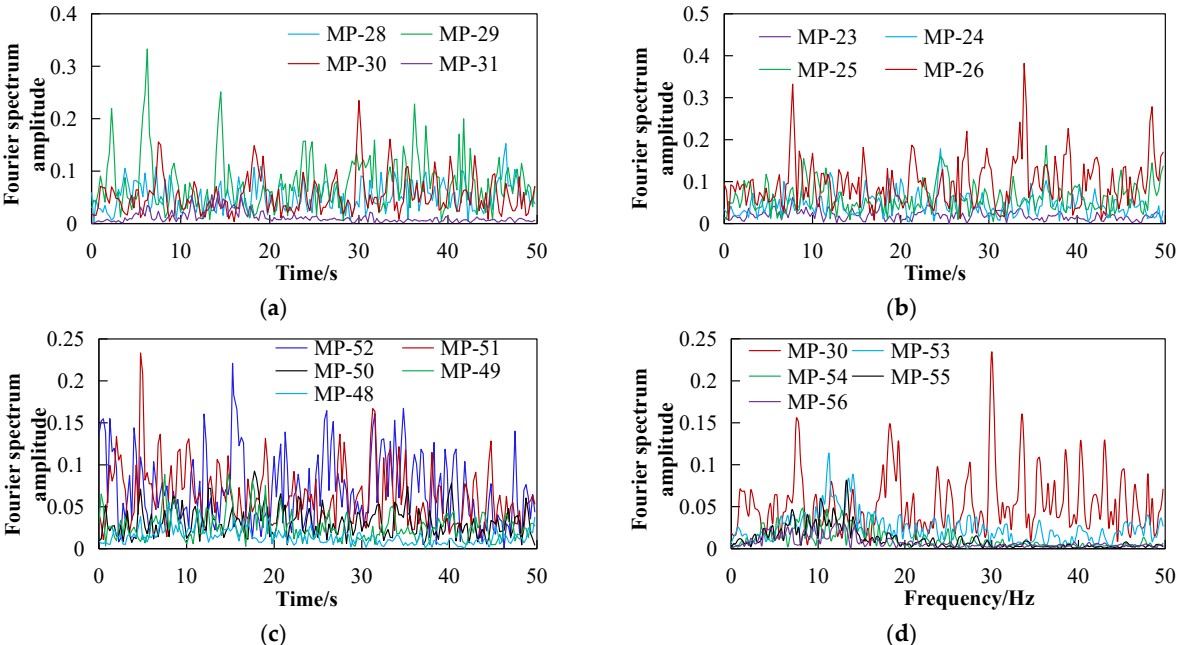

**Figure 16.** Fourier spectrum of horizontal acceleration wavelet excitation from the explosion source: (**a**) Right side of the explosion source; (**b**) Left of the source of the explosion; (**c**) Below the source of the explosion; (**d**) At the slope surface.

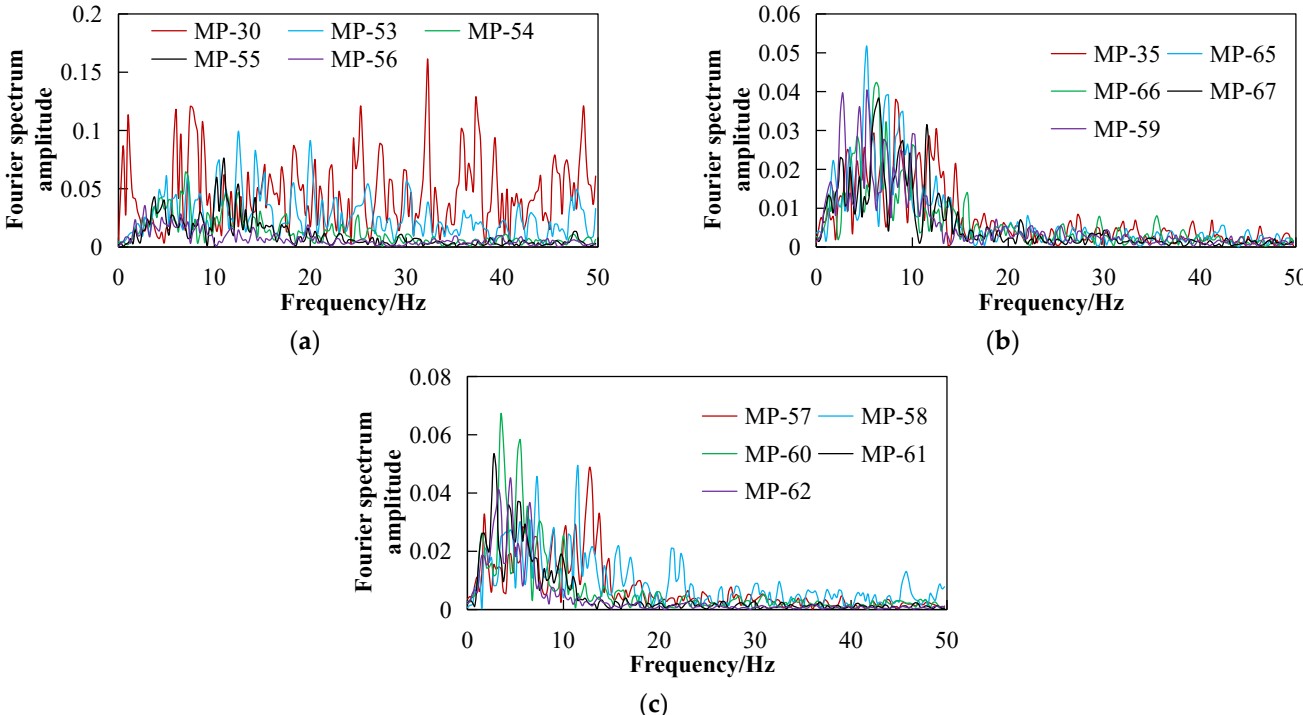

**Figure 17.** Fourier spectrum of vertical acceleration wavelet excitation from the explosion source at the slope surface: (**a**) slope surface; (**b**) inside the slope; (**c**) slope crest.

### 3.2.2. Attenuation Law of Wave Triggered by Explosion Source

To study the dynamic response characteristics of S- and P-waves induced by the explosion source, the peak Fourier spectrum amplitude (PFSA) of acceleration-time histories is selected as the analysis index, and the PFSA changes of typical measuring points of the model are studied. In different regions of the DEM model, the ratio of PFSA$z$/PFSA$x$ of seismic waves generated by the explosion source are shown in Figure 18. PFSA$x$ and PFSA$z$ are PFSA of Fourier spectra of S-wave and P-wave excited by the explosion source, respectively. Figure 18a shows that the PFSA$z$/PFSA$x$ in the slope region is 0.7–0.85 overall, indicating that the dynamic response of the S-wave that stimulates seismic waves in the slope region is greater than that of the P-wave. Figure 18b–d shows that PFSA$z$/PFSA$x$ of the transverse and vertical axes of MP-27 is 1.15–1.8, overall, indicating that the dynamic response of S-wave that generates seismic waves in the bedrock region is smaller than that of P-wave. PFSA changes and distribution of typical measuring points in the model are shown in Figures 19 and 20. Figure 19a shows that the PFSA on the slope surface decreases with the increase in elevation. However, the variation law of PFSA inside the slope is opposite to that of the slope surface, showing a slow increase trend with the slope elevation. This indicates that the seismic wave has an attenuation effect on the slope surface but has an obvious elevation amplification effect inside the slope. At the same time, the PFSA in the slope is smaller than that on the slope surface, which indicates that the amplifying effect of seismic waves on the slope surface is greater than that of the sloping interior, that is, the slope has an attenuation effect on the energy propagation of seismic waves. Figure 19b,c and Figure 20 show that, with the increase in the distance from the explosion source, the PFSA of seismic waves gradually decreases. In particular, the PFSA is relatively large within the range of 25 m from the explosion source, while the PFSA is relatively small in the area of 100 m away from the explosion source. The dynamic response of the seismic wave generated by the explosion source is larger within certain a distance (<25 m). When the distance from the source is greater than 100 m, the seismic wave energy attenuates rapidly, and the dynamic response of the seismic wave in the remote bedrock area is small. In other words, the explosion source has a limited range of influence, which has the greatest influence on the dynamic response within 25 m away from the explosion source, and less influence on the area more than 100 m away from the explosion source.

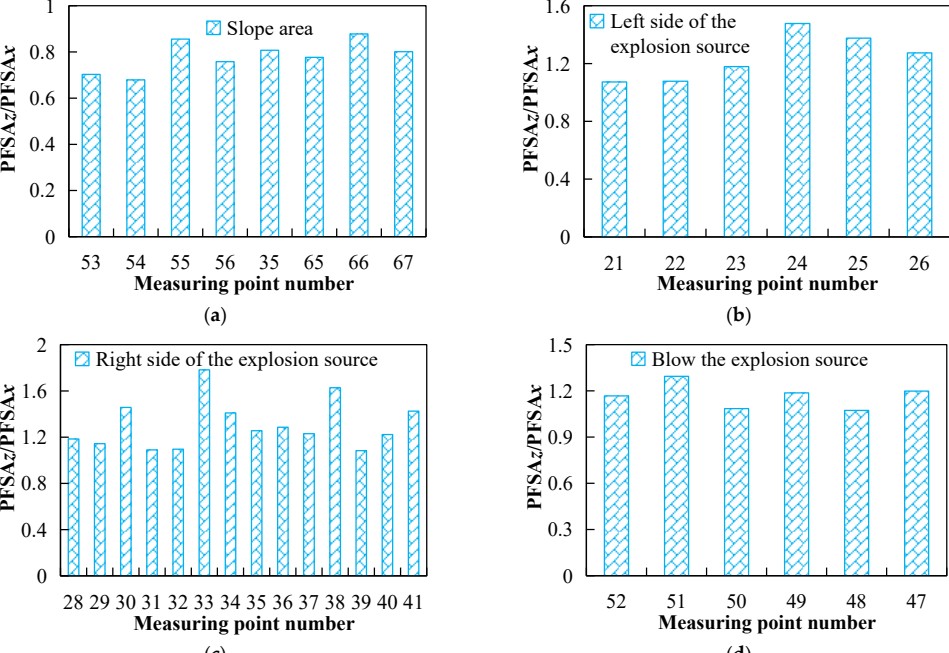

**Figure 18.** The ratio of PFSA$x$ to PFSA$z$ in the DEM model: (**a**) Slope area; (**b**) Left side of the explosion source; (**c**) Right side of the explosion source; (**d**) Blow the explosion source.

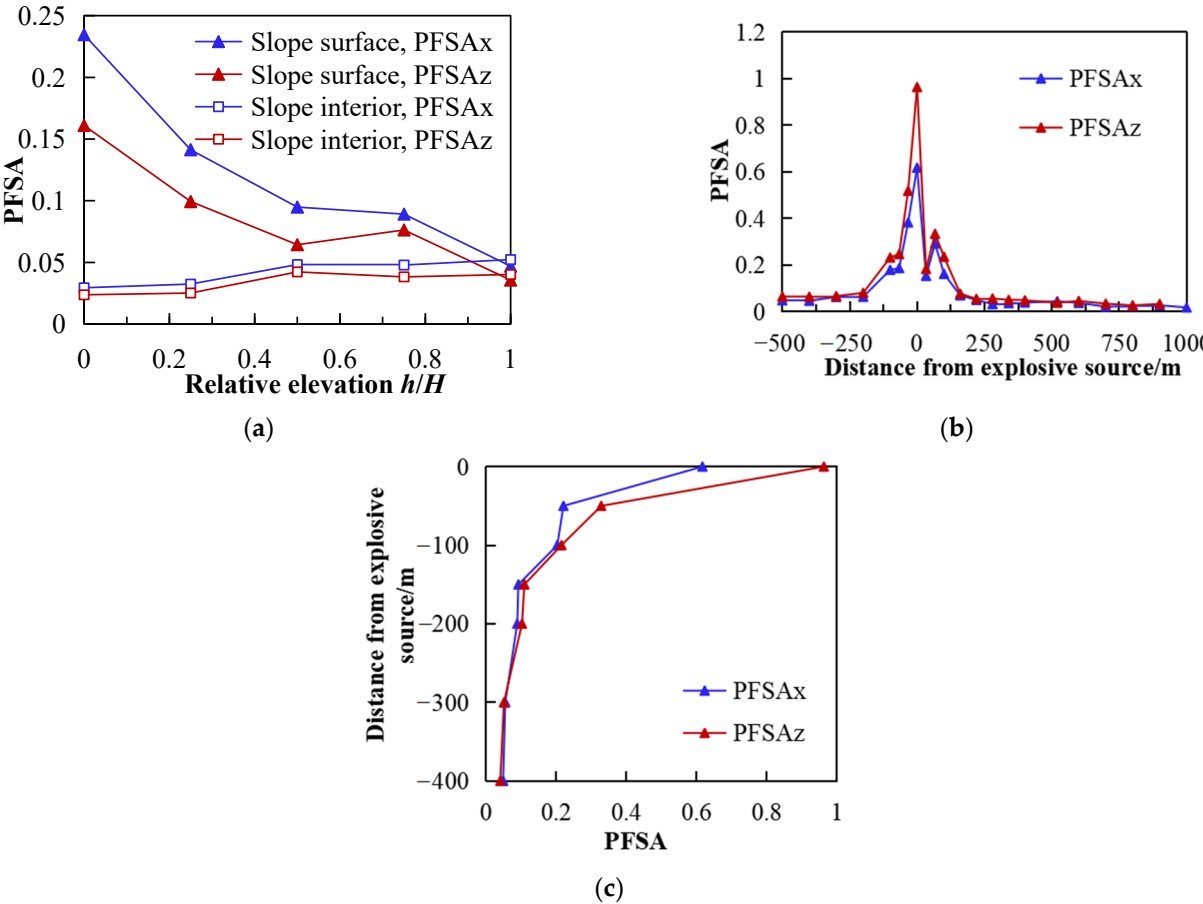

**Figure 19.** Change of PFSA in the DEM model: (**a**) in the slope area; (**b**) in the horizontal axial direction of the explosion source; (**c**) in the vertical axial direction of the explosion source.

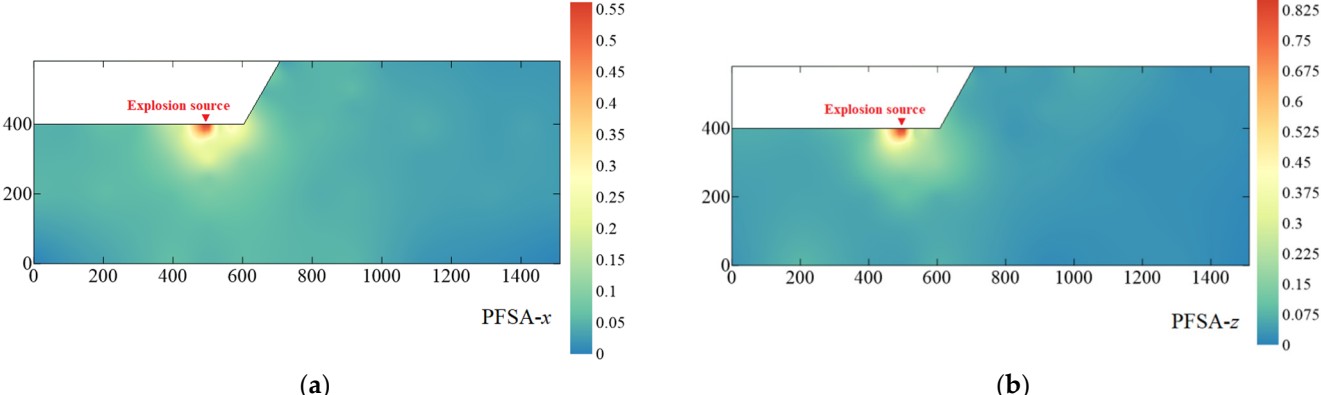

**Figure 20.** Distribution of peak Fourier spectrum amplitude (PFSA) of acceleration wavelet excitation from the explosion source: (**a**) PFSAx; (**b**) PFSAz.

## 4. Attenuation Law of Wave Induced by Explosion Source Using Stress and Energy-Based Method

To further clarify the dynamic response characteristics and attenuation law of seismic waves triggered by explosion sources from the perspective of stress and energy, the stress distribution characteristics and the normal force on each boundary of the model are shown in Figure 21. Figure 21a shows that the stress in the model is small overall ($<3.0 \times 10^7$), but it can be seen that there are some stress zones with large values ($15 \sim 20 \times 10^7$) towards the bottom of the bedrock. Meanwhile, Figure 21b shows that the maximum normal force is found at the bottom boundary of the model bedrock. This indicates that the seismic

waves induced by the blasting source have a greater impact on the bottom boundary of the bedrock than other boundaries of the model. In other words, compared with the boundary of the left and right sides of the model, the propagation of blasting wave to the deep bedrock attenuates more slowly, and the normal force at the bottom boundary is the largest. In addition, the time-history curve of blasting excitation seismic wave energy and heat generation in the model is shown in Figure 22. Figure 22a shows that during the blasting process, the energy of excited seismic waves decreases rapidly within 0–0.03 s. When the blasting time lasts for 0.3 s, the seismic wave energy gradually becomes stable. Figure 22b shows that the heat generated by the model increases rapidly within 0–0.3 s under the blasting action and increases slowly, gradually becoming stable after 0.3 s of blasting time. It can be seen that the blasting excitation seismic wave energy has a rapid attenuation effect within a short time (0–0.3 s), and the attenuation effect gradually decreases and gradually becomes stable when the blasting time is >0.3 s.

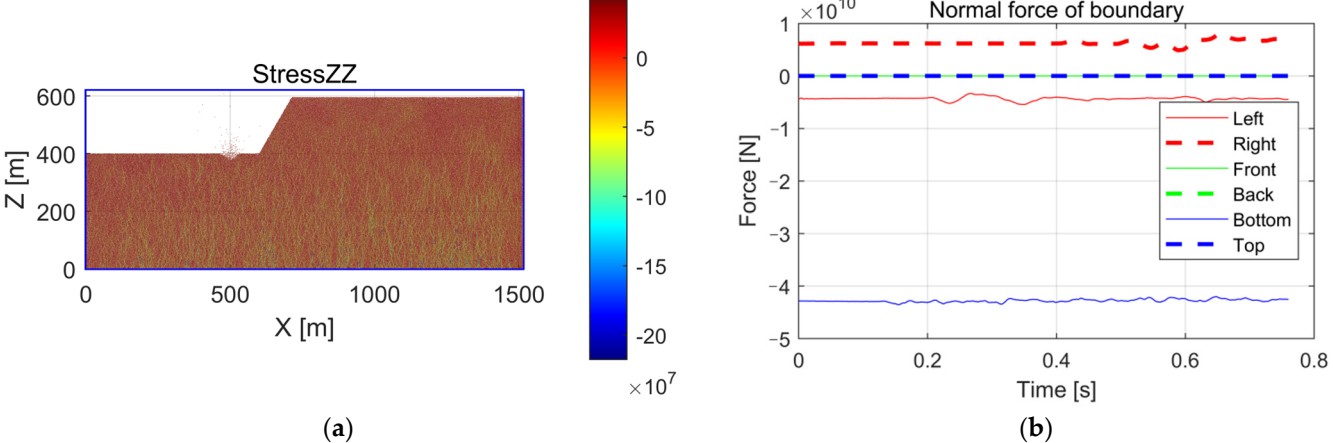

**Figure 21.** Stress and normal force of boundary of the DEM model under explosion: (**a**) distribution of stress; (**b**) curves of normal force of boundary.

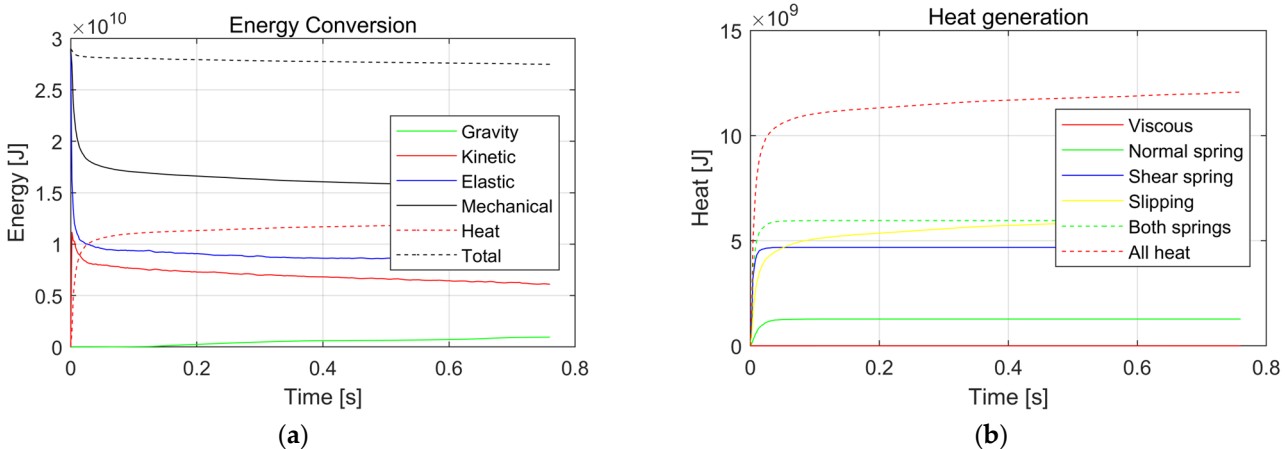

**Figure 22.** Energy and heat curve of the DEM model under explosion: (**a**) Energy-time history curves; (**b**) Heat-time history curves.

## 5. Discussion

In this work, the seismic wave propagation characteristics of granite slope sites under blasting are studied by MatDEM. From the perspective of time and frequency domains, the dynamic response characteristics and attenuation law of seismic waves in slope and bedrock areas are discussed from the perspective of multi-parameter and multi-angle. The results of time-domain analysis mainly show the propagation characteristics of seismic waves under blasting from the perspective of mechanical deformation. The frequency-

domain analysis mainly reveals the propagation characteristics of seismic waves from the perspective of inherent characteristics. The analysis results in the time and frequency domains are complementary to each other. In addition, based on MatDEM numerical simulation, the dynamic response characteristics of blasting-induced seismic waves can be systematically analyzed from the perspective of the velocity field, displacement field, and energy transmission. This work can provide an important reference for the dynamic response of blasting-induced seismic waves in rocky sites. However, in this work, MatDEM is used to study the propagation characteristics of explosion-induced seismic waves in a granite site, which is a special case. For other rocky sites, the wave propagation characteristics of explosion-induced seismic waves need to be discussed, and further numerical simulation of sites with different lithologies are also needed. At the same time, this study also needs to use field tests and laboratory model tests for further verification, which will be the focus of the next research.

## 6. Conclusions

MatDEM is used to study the wave propagation characteristics and attenuation law of seismic waves triggered by explosion sources in the rock site. Some main conclusions can be drawn as follows:

1. The analysis of the velocity and displacement fields of waves shows that the wave propagation in the rock slope site is characterized by the arc. With the explosion duration and the increase in the distance from the explosion source, the amplitude of waves attenuated gradually. The wave attenuation effect of the ground surface is smaller than that in the bedrock. Near the explosion source (<25 m), the PGA of waves on the right side of the explosion source is the largest but decays rapidly, while the PGA on the left and lower side of the explosion source is smaller while the PGA on the lower side is the smallest.

2. Time and frequency domain analysis shows that the waves attenuate gradually with the increase in the distance from the explosion source. The explosion source has the greatest influence on the dynamic response within the range of 25 m, and the influence is small when it is beyond 100 m from the explosion source. The dynamic response of waves is larger in the range of 25 m from the explosion source, and the attenuation rate is the fastest. The attenuation rate is small and tends to be stable in the area of 100 m away from the explosion source. The slope region can weaken the dynamic response of waves, especially filtering the seismic waves of the high-frequency band (>20 Hz), which greatly reduces the amplitude of the high-frequency spectrum. The seismic wave energy has a rapid attenuation effect within a short time (0–0.3 s), and the attenuation effect gradually decreases and becomes stable when the blasting time is >0.3 s.

3. The existence of a slope changes the dynamic response characteristics of waves. In the bedrock and the ground surface, the dynamic response of the P-wave is greater than that of the S-wave, and the PGV and PGA of the P-wave are greater than that of the S-wave. The PGV and PGA of the S-wave in the slope region are smaller than those of the P-wave, and the dynamic amplification effect of the P-wave is larger than that of the S-wave. The waves in the slope region show an attenuation effect along the slope surface. The PGV and PGA of waves in the slope interior increase along with the increase in elevation, which has a typical elevation amplification effect.

**Author Contributions:** D.S.: conceptualization, methodology, writing—original draft. D.S.: data curation, software. M.L. and C.L.: writing—review and editing. X.Q.: supervision, writing—review. M.L. and C.L.: funding acquisition, resources, validation, writing—review and editing. D.S.: supervision, writing—review. W.L., X.W. and D.H.: editing. All authors have read and agreed to the published version of the manuscript.

**Funding:** This study was funded from the supported of the National Natural Science Foundation of China (No.s 52109125 and 52090081), the Independent research project of State key Laboratory of Hydro science and Engineering (No. 2022-KY-02), the Open Research Fund of SINOPEC Key Laboratory of Geophysics (No. WX2021-01-12), the China Postdoctoral Science Foundation (No. 2020M680583), the National Postdoctoral Program for Innovative Talent of China (No. BX20200191), and the Natural Science Foundation of Jiangsu Province (No. BK20130481).

**Institutional Review Board Statement:** Not applicable.

**Informed Consent Statement:** Not applicable.

**Data Availability Statement:** Not applicable.

**Conflicts of Interest:** The authors declare that they have no known competing financial interests or personal relationships that could have appeared to influence the work reported in this paper.

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
