# Peer review of "Investigation on the Seismic Wave Propagation Characteristics Excited by Explosion Source in High-Steep Rock Slope Site Using Discrete Element Method"

_sustainability, doi:10.3390/su142417028_

Round 1

Reviewer 1 Report

Please see my comments in the attachments.

Author Response

Dear Editors and Reviewers:

Thank you for your letter and for the comments concerning our manuscript entitled “Investigation on the seismic wave propagation characteristics excited by explosion source in high-steep rock slope site using discrete element method”. The comments were valuable and helpful for revising and improving our paper and clarifying the significance of our research. We have thoroughly amended the paper according to your comments and answered the technical questions with point-by-point responses. We hope our corrections will be met with your approval. The revised portions are marked in the paper. The main corrections in the paper and the responses to the editors’ and reviewers’ comments are as follows:

Comments from the reviewers:

-Reviewer 1

Sincerely thank you for your careful reading and revision of the manuscript. Regarding the comments and suggestions proposed in the review, we made the following changes.

Comment: (The paper titled "Investigation on the seismic wave propagation characteristics excited by explosion source in high-steep rock slope site using discrete element method" by Song et al. is focused on an interesting topic. The influence of seismic waves induced by explosion sources on the dynamic response characteristics of rock slope sites is one of the important problems affecting engineering construction. In this paper, the high performance matrix discrete element method (MatDEM) is used to carry out numerical simulation tests on the granite rock medium site. The means used in this study are reasonable and necessary, with several outcomes obtained. I suggest that this paper can be accepted after minor revision. However, there are still some issues that should be clarified and complemented before publication.)

  1. Comment:(Abstract should be brief and concise. The innovation and main research results of this work should be fully displayed. Please emphasize the innovation of this paper.)

Response: We have made corrections according to the Reviewer’s comments. We have rewritten the abstract, as follows.

Abstract: The influence of seismic waves induced by explosion sources on the dynamic response characteristics of rock slope sites is one of the important problems affecting engineering construction. To investigate the wave propagation characteristics and attenuation law of seismic waves induced by explosive sources in rock site from the perspective of time and frequency domains, the high performance matrix discrete element method (MatDEM) is used to carry out numerical simulation tests on a granite rock medium site. The discrete element model of the high-steep rock slope is established by MatDEM, and the dynamic analysis of the rock medium site is carried out by loading blasting vibration load to generate seismic waves. The results show that the seismic waves in the rock site present the characteristics of arc propagation attenuation. The maximum attenuation rate of the dynamic response is the fastest within 0.3 s and 25 m from the explosion source. The slope region can weaken the dynamic response of seismic waves generated by the explosion source, in particular, the high-frequency band (>20 Hz) has an obvious filtering effect. The dynamic response of the P-wave induced by the explosive source is greater than that of the S-wave in the bedrock and surface region. The dynamic amplification effect of the P-wave is greater than that of the S-wave in the slope region. The seismic waves in the slope region show an attenuation effect along the slope surface and have a typical elevation amplification effect inside the slope.”

  1. Comment:(This paper seen to be too long. Please simplify the content of this paper as much as possible to enhance its logic and readability.)

Response: We have made corrections according to the Reviewer’s comments. We have simplified the content of this paper as much as possible to enhance its logic and readability.

  1. Comment:(MatDEM has been applied in the field of slope engineering. Please add an overview of this aspect in the introduction. This is beneficial for readers to understand MatDEM software.)

Response: We have made corrections according to the Reviewer’s comments.

“Moreover, MatDEM has been applied to geotechnical engineering by some scholars, in particular, the slope disaster process analysis has been discussed via MatDEM. Chen and Song used MatDEM to study the evolution process and failure mode of loess bedrock landslide [18]. Chen et al. used MatDEM to simulate the failure evolution process of reservoir bank landslide and forecast the danger areas affected by reservoir bank landslide [19]. Li et al. Used MatDEM to study the deformation and failure process and evolution of fractured rock mass [20]. ”

  1. 4. Comment:(The font in Figure 4 is a little small. Please enlarge it.)

Response: We have made corrections according to the Reviewer’s comments.

  1. Comment:(Please rewrite the conclusion section. Please list the main conclusions of this study to highlight the innovation of this study.)

Response: We have made corrections according to the Reviewer’s comments. We have rewritten the conclusions.

Conclusions

MatDEM is used to study the wave propagation characteristics and attenuation law of seismic waves triggered by explosion sources in the rock site. Some main conclusions can be drawn as follows:

  1. The analysis of the velocity and displacement fieldsofwaves shows that the wave propagation in the rock slope site is characterized by the arc. With the explosion duration and the increase of the distance from the explosion source, the amplitude of waves attenuated gradually. The wave attenuation effect of the ground surface is smaller than that in the bedrock. Near the explosion source (<25 m), the PGA of waves on the right side of the explosion source is the largest but decays rapidly, while the PGA on the left and lower side of the explosion source is smaller, in particular, the PGA on the lower side is the smallest.
  2. Timeand frequency domainanalysis shows that the waves attenuate gradually with the increase of the distance from the explosion source. The explosion source has the greatest influence on the dynamic response within the range of 25 m, and the influence is small when it is beyond 100 m from the explosion source. The dynamic response of waves is larger in the range of 25 m from the explosion source, and the attenuation rate is the fastest. The attenuation rate is small and tends to be stable in the area of 100 m away from the explosion source. The slope region can weaken the dynamic response of waves, especially filtering the seismic waves of the high-frequency band (>20 Hz), which greatly reduces the amplitude of the high-frequency spectrum. The seismic wave energy has a rapid attenuation effect within a short time (0-0.3 s), and the attenuation effect gradually decreases and becomes stable when the blasting time is >0.3 s.
  3. The existence of a slope changes the dynamic response characteristics ofwaves.In the bedrock and the ground surface, the dynamic response of the P-wave is greater than that of the S-wave, and the PGV and PGA of the P-wave are greater than that of the S-wave. The PGV and PGA of the S-wave in the slope region are smaller than those of the P-wave, and the dynamic amplification effect of the P-wave is larger than that of the S-wave. The waves in the slope region show an attenuation effect along the slope surface. The PGV and PGA of waves in the slope interior increase along the increase of elevation, which has a typical elevation amplification effect.”
  4. 6. Comment:(Please add a discussion section to clearly point out the limitations and novelty of this study. This is very important for readers to fully understand this paper. Please add this part.)

Response: We have made corrections according to the Reviewer’s comments.

6. Discussions

In this work, the seismic wave propagation characteristics of granite slope site under blasting are studied by MatDEM. From the perspective of time and frequency domains, the dynamic response characteristics and attenuation law of seismic waves in slope and bedrock areas are discussed from the perspective of multi-parameter and multi-angle. The result of time-domain analysis mainly shows the propagation characteristics of seismic waves under blasting from the perspective of mechanical deformation. The frequency-domain analysis mainly reveals the propagation characteristics of seismic waves from the perspective of inherent characteristics. The analysis results in the time and frequency domains are complementary to each other. In addition, based on MatDEM numerical simulation, the dynamic response characteristics of blasting-induced seismic waves can be systematically analyzed from the perspective of the velocity field, displacement field, and energy transmission. This work can provide an important reference for the dynamic response of blasting-induced seismic waves in rocky sites. However, in this work, MatDEM was used to study the propagation characteristics of explosion-induced seismic waves in a granite site, which is a special case. For other rocky sites, the wave propagation characteristics of explosion-induced seismic waves need to be discussed, and further numerical simulation of sites with different lithologies is also needed. At the same time, this study also needs to use field tests and laboratory model tests for further verification, which will be the focus of the next research.”

  1. Comment:(The English writing of this paper needs to be improved. Please check the grammar errors and improper expressions in this paper.)

Response: We have made corrections according to the Reviewer’s comments. We have checked the grammar errors and improper expressions in this paper.

Special thanks to you for your helpful comments.

We tried our best to improve the manuscript and made some changes to the manuscript. These changes do not influence the content or framework of the paper. We did not list here all of the changes, but we marked them in the revised manuscript. We appreciate the work of the Editor and Reviewers and hope that the corrections will be met with your approval. Once again, thank you very much for your comments and suggestions.

Yours Sincerely,

Prof. Menxin Liu

School of Civil Engineering, Northeast Forestry University, Harbin 150040, China

Corresponding author. E-mail address: liumengxin@nefu.edu.cn (Mengxin Liu)

Prof. Chun Liu

School of Earth Sciences and Engineering, Nanjing University, Nanjing 210023, China

Corresponding author. E-mail address: chunliu@nju.edu.cn (Chun Liu)

Reviewer 2 Report

Song et al. used high performance matrix discrete element method (MatDEM) to study the propagation characteristics of seismic waves induced by explosion source in the rock slope site. It is helpful for exploration and engineering disaster reduction to investigate the propagation characteristics of excited seismic wave in rock mass under blasting action. Blasting load is a common factor in engineering construction. This research adopts a new discrete element method and obtains some interesting results. On the whole, the research work of this paper is good. I suggest that this paper can be accepted after revision. However, the following suggestions should be carefully revised before publication. The specific modification suggestions are as follows.

1. In 2. Methodology, the application of blasting load is unclear, please further clarify it.

2. Please further clarify the meaning of PGVz/PGVx and PGAz/PGAx in Figure 11. The current form makes readers puzzled.

3. The size of Figure 4a does not seem to correspond to Figure 4b. Please check and modify it.

4. The blasting load has adverse effects on engineering rock mass slope. It is better to add some photos of rock-soil mass instability caused by blasting load, so that readers can intuitively understand the catastrophic effect of blasting load.

5. Please add an introduction to the novelty of the MatDEM discrete element method used in this work, and also clarify the difference from other discrete element methods.

6. The abstract of this paper should be further revised to emphasize the innovation of the research.

7. The formula in the paper looks ugly, please modify them.

8. The conclusion section needs to be further refined, please revise it.

9. It is suggested to discuss the advantages and disadvantages of this work, which will help readers understand the research value of this work. At the same time, it is suggested that the research results of this work should be verified in the future research work.

Author Response

Dear Editors and Reviewers:

Thank you for your letter and for the comments concerning our manuscript entitled “Investigation on the seismic wave propagation characteristics excited by explosion source in high-steep rock slope site using discrete element method”. The comments were valuable and helpful for revising and improving our paper and clarifying the significance of our research. We have thoroughly amended the paper according to your comments and answered the technical questions with point-by-point responses. We hope our corrections will be met with your approval. The revised portions are marked in the paper. The main corrections in the paper and the responses to the editors’ and reviewers’ comments are as follows:

Comments from the reviewers:

-Reviewer 2

Sincerely thank you for your careful reading and revision of the manuscript. Regarding the comments and suggestions proposed in the review, we made the following changes.

Comment: (Song et al. used high performance matrix discrete element method (MatDEM) to study the propagation characteristics of seismic waves induced by explosion source in the rock slope site. It is helpful for exploration and engineering disaster reduction to investigate the propagation characteristics of excited seismic wave in rock mass under blasting action. Blasting load is a common factor in engineering construction. This research adopts a new discrete element method and obtains some interesting results. On the whole, the research work of this paper is good. I suggest that this paper can be accepted after revision. However, the following suggestions should be carefully revised before publication. The specific modification suggestions are as follows.)

  1. Comment:(In 2. Methodology, the application of blasting load is unclear, please further clarify it.)

Response: We have made corrections according to the Reviewer’s comments.

“Blasting load dynamite of 2172.67 kg is applied to MP-27 (explosion source) for blasting to excite seismic waves.”

  1. Comment:(Please further clarify the meaning of PGVz/PGVx and PGAz/PGAx in Figure 11. The current form makes readers puzzled.)

Response: We have made corrections according to the Reviewer’s comments.

“In addition, to further explore the dynamic response characteristics and attenuation rules of S- and P-waves, the PGVx/PGVz and PGAx/PGAz in different regions of the model are shown in Fig. 11. PGVx and PGAx are the PGV and PGA of S-wave, and PGVz and PGAz are PGV and PGA of P-wave, respectively.”

  1. Comment:(The size of Figure 4a does not seem to correspond to Figure 4b. Please check and modify it.)

Response: We have made corrections according to the Reviewer’s comments.

(a)                                  (b)

(c)

Figure 4. DEM model: (a) Model box; (b) DEM model (MatDEM) for the slope; (c) layout of measuring points.

  1. Comment:(The blasting load has adverse effects on engineering rock mass slope. It is better to add some photos of rock-soil mass instability caused by blasting load, so that readers can intuitively understand the catastrophic effect of blasting load.)

Response: We have made corrections according to the Reviewer’s comments.

Because there are too many pictures and text on the paper, some photos are not added to the paper. The pictures of rock mass rupture caused by blasting are as follows.

  1. Comment:(Please add an introduction to the novelty of the MatDEM discrete element method used in this work, and also clarify the difference from other discrete element methods.)

Response: We have made corrections according to the Reviewer’s comments.

“Moreover, MatDEM has been applied to geotechnical engineering by some scholars, in particular, the slope disaster process analysis has been discussed via MatDEM. Chen and Song used MatDEM to study the evolution process and failure mode of loess bedrock landslide [18]. Chen et al. used MatDEM to simulate the failure evolution process of reservoir bank landslide and forecast the danger areas affected by reservoir bank landslide [19]. Li et al. Used MatDEM to study the deformation and failure process and evolution of fractured rock mass [20]. ”

  1. Comment:(The abstract of this paper should be further revised to emphasize the innovation of the research.)

Response: We have made corrections according to the Reviewer’s comments. We have rewritten the abstract, as follows.

Abstract: The influence of seismic waves induced by explosion sources on the dynamic response characteristics of rock slope sites is one of the important problems affecting engineering construction. To investigate the wave propagation characteristics and attenuation law of seismic waves induced by explosive sources in rock sites from the perspective of time and frequency domains, the high performance matrix discrete element method (MatDEM) is used to carry out numerical simulation tests on a granite rock medium site. The discrete element model of the high-steep rock slope is established by MatDEM, and the dynamic analysis of the rock medium site is carried out by loading blasting vibration load to generate seismic waves. The results show that the seismic waves in the rock site present the characteristics of arc propagation attenuation. The maximum attenuation rate of the dynamic response is the fastest within 0.3 s and 25 m from the explosion source. The slope region can weaken the dynamic response of seismic waves generated by the explosion source, in particular, the high-frequency band (>20 Hz) has an obvious filtering effect. The dynamic response of the P-wave induced by the explosive source is greater than that of the S-wave in the bedrock and surface region. The dynamic amplification effect of the P-wave is greater than that of the S-wave in the slope region. The seismic waves in the slope region show an attenuation effect along the slope surface and have a typical elevation amplification effect inside the slope.”

  1. Comment:(The formula in the paper looks ugly, please modify them.)

Response: We have made corrections according to the Reviewer’s comments. Please check them.

  1. Comment:(The conclusion section needs to be further refined, please revise it.)

Response: We have made corrections according to the Reviewer’s comments. We have rewritten the conclusions.

“5. Conclusions

MatDEM is used to study the wave propagation characteristics and attenuation law of seismic waves triggered by explosion sources in the rock site. Some main conclusions can be drawn as follows:

  1. The analysis of the velocity and displacement fieldsofwaves shows that the wave propagation in the rock slope site is characterized by the arc. With the explosion duration and the increase of the distance from the explosion source, the amplitude of waves attenuated gradually. The wave attenuation effect of the ground surface is smaller than that in the bedrock. Near the explosion source (<25 m), the PGA of waves on the right side of the explosion source is the largest but decays rapidly, while the PGA on the left and lower side of the explosion source is smaller, in particular, the PGA on the lower side is the smallest.
  2. Timeand frequency domainanalysis shows that the waves attenuate gradually with the increase of the distance from the explosion source. The explosion source has the greatest influence on the dynamic response within the range of 25 m, and the influence is small when it is beyond 100 m from the explosion source. The dynamic response of waves is larger in the range of 25 m from the explosion source, and the attenuation rate is the fastest. The attenuation rate is small and tends to be stable in the area of 100 m away from the explosion source. The slope region can weaken the dynamic response of waves, especially filtering the seismic waves of the high-frequency band (>20 Hz), which greatly reduces the amplitude of the high-frequency spectrum. The seismic wave energy has a rapid attenuation effect within a short time (0-0.3 s), and the attenuation effect gradually decreases and becomes stable when the blasting time is >0.3 s.
  3. The existence of slope changes the dynamic response characteristics ofwaves.In the bedrock and the ground surface, the dynamic response of the P-wave is greater than that of the S-wave, and the PGV and PGA of the P-wave are greater than that of the S-wave. The PGV and PGA of the S-wave in the slope region are smaller than those of the P-wave, and the dynamic amplification effect of the P-wave is larger than that of the S-wave. The waves in the slope region show an attenuation effect along the slope surface. The PGV and PGA of waves in the slope interior increase along the increase of elevation, which has a typical elevation amplification effect.
  4. Comment:(It is suggested to discuss the advantages and disadvantages of this work, which will help readers understand the research value of this work. At the same time, it is suggested that the research results of this work should be verified in the future research work.)

Response: We have made corrections according to the Reviewer’s comments.

4. Discussions

In this work, the seismic wave propagation characteristics of granite slope sites under blasting are studied by MatDEM. From the perspective of time and frequency domains, the dynamic response characteristics and attenuation law of seismic wave in slope and bedrock area are discussed from the perspective of multi-parameter and multi-angle. The result of time-domain analysis mainly shows the propagation characteristics of seismic waves under blasting from the perspective of mechanical deformation. The frequency-domain analysis mainly reveals the propagation characteristics of seismic waves from the perspective of inherent characteristics. The analysis results in the time and frequency domains are complementary to each other. In addition, based on MatDEM numerical simulation, the dynamic response characteristics of blasting-induced seismic waves can be systematically analyzed from the perspective of the velocity field, displacement field, and energy transmission. This work can provide an important reference for the dynamic response of blasting-induced seismic waves in rocky sites. However, in this work, MatDEM was used to study the propagation characteristics of explosion-induced seismic waves in a granite site, which is a special case. For other rocky sites, the wave propagation characteristics of explosion-induced seismic waves need to be discussed, and further numerical simulation of sites with different lithologies is also needed. At the same time, this study also needs to use field tests and laboratory model tests for further verification, which will be the focus of the next research.”

Special thanks to you for your helpful comments.

We tried our best to improve the manuscript and made some changes to the manuscript. These changes do not influence the content or framework of the paper. We did not list here all of the changes, but we marked them in the revised manuscript. We appreciate the work of the Editor and Reviewers and hope that the corrections will be met with your approval. Once again, thank you very much for your comments and suggestions.

Yours Sincerely,

Prof. Menxin Liu

School of Civil Engineering, Northeast Forestry University, Harbin 150040, China

Corresponding author. E-mail address: liumengxin@nefu.edu.cn (Mengxin Liu)

Prof. Chun Liu

School of Earth Sciences and Engineering, Nanjing University, Nanjing 210023, China

Corresponding author. E-mail address: chunliu@nju.edu.cn (Chun Liu)

Reviewer 3 Report

Specific comments to the Authors

The manuscript entitled “Investigation on the seismic wave propagation characteristics excited by explosion source in high-steep rock slope site using discrete element method” presents an interesting study that may provide scientific knowledge regarding the dynamic response of rock slope sites to seismic waves produced by explosive material for rock breaking during tunnel construction. The study specifically focuses on determining the way different points of a granite site react to seismic waves produces by a blast close to the slope. The manuscript is well written, although a review of the English language is recommended, since several grammar and syntax issues were detected. The structure of the text is well organized, although small reconfiguration is recommended. Figures and tables are well displayed and sufficiently described to support the conducted survey, with few minor exceptions that have been pointed in the specific comments to the authors. The bibliography is up to date and the appropriate citations exceed where needed, with few minor exceptions that have been pointed in the specific comments to the authors. There are several corrections and clarifications that have to be made prior to publication in the Sustainability Journal, therefore Minor Revision is suggested.  

(1) General comment: Please separate the chapters by adding main Chapter at the beginning. For example for Chapter “3. Wave propagation characteristics of seismic waves triggered by explosion source in the time domain” add first Chapter 3 Results and after the rest “3. Wave propagation characteristics of seismic waves triggered by explosion source in the time domain”.  Please apply the same for Chapter 4 Discussion, for Chapter 5 Conclusions …  and so on.

Abstract

(2) Line 16 : “ … construction ”. Please add a sentence after “construction” stating what the goal of the study is. 

(3) Line 25 : “The … to Line 29 … region.” Please break the long sentences into shorter ones that have one meaning. Please apply the same for the rest of the manuscript.

Introduction

(4) General comment: The Introduction is a bit too long. Please make it shorter. In addition, a review of the English language is recommended, since several grammar and syntax issues were detected.

(5) Line 55: “ Many scholars … to line 73 … and so on”. Please do not include a list of what each scholar has done. Either remove the entire paragraph or try to combine the work that have done and how does that connect or add to your study. This segment should be included in the Discussion Chapter.

(6) Line 96 Wu et al. … to Line 102 … geological conditions. Same comment as for comment (4)

(7) Line 111: For Figure 1 there is citation [33] however in the context Line 86 the citation is [25]. Please add the right citation in both places.

(8) Line 128 : Please add after “…application value” 2 sentences, one stating what is the main goal of the study and one stating what are the main objectives.

Methodology

Principle of MatDEM

(9) General comment: Please make a more clear definition and elaboration of the methods for this study by separating them in distinct chapters. For example, use a chapter to present the “Fast GPU Matrix computing of the Discrete Element Method (MatDEM)” another chapter to present The discrete element numerical model of high-steep rock slope” …  “Discrete element dynamic analysis of the rock medium” “The time-frequency analysis method” and so on …

Wave propagation characteristics of seismic waves triggered by explosion source in the time domain

Propagation characteristics of the seismic wave field

(10) Line 297: “Figure 6”. Figure 6 is the same as figure 5. Please remove figures 6 and add the right ones, if applicable.

Waveform characteristics of waves triggered by explosion source

(11) Line 303 : Figure 7 must be placed after Chapter 3.2 Waveform characteristics of waves triggered by explosion source. In addition, consider presenting the figures in this Chapter based on the different points MP-27 only … and so on, rather showing them together as it looks cluttered and it is difficult to follow the results presented in the text in the figures. If decided to change the formatting of the figures please apply for the entire manuscript.

Dynamic response characteristics of S- and P-waves of seismic wave triggered by explosion source 

(12) Line 384: “It can … to Line 387 … P-wave”. Please move this section to Discussion as this part is supposed to be Results where there is only presentation of the results without explanation.

(13) General comment : Please try to put the Figures used in the manuscript closer to the text where they are mentioned. For example Line 419 mentions “Fig.15.” which can be found almost after 2 pages. Please try to bring it closer. Same with the rest of the Figures used in the manuscript.

(14) General comment : Most of Chapters 4 and 5 are results and should be included with Chapter 3 in one single Chapter, Results. In addition, there is no Discussion Chapter which is necessary in order for a manuscript to be published. Discussion is probably the most important Chapter from a publication as it elaborates on the findings, how do they connect to previous studies and what is the major contribution of this study to the Engineering and Construction Sectors.  For example read comment 5, in addition Line 480 “ This indicates … to line 484 … as an example.” should be part of Discussion. Also Line 384 “It can … to line 386 … waves” this is also Discussion.

(15) General comment : Most of the things included in Chapter 6 Conclusions are not actual conclusions but more so Results. The conclusions come out of the Discussion Chapter. Since there is no Discussion Chapter there cannot be any Conclusions. So please write a Discussion Chapter first and after that write the Conclusions Chapter.

Author Response

Dear Editors and Reviewers:

Thank you for your letter and for the comments concerning our manuscript entitled “Investigation on the seismic wave propagation characteristics excited by explosion source in high-steep rock slope site using discrete element method”. The comments were valuable and helpful for revising and improving our paper and clarifying the significance of our research. We have thoroughly amended the paper according to your comments and answered the technical questions with point-by-point responses. We hope our corrections will be met with your approval. The revised portions are marked in the paper. The main corrections in the paper and the responses to the editors’ and reviewers’ comments are as follows:

Comments from the reviewers:

-Reviewer 3

Sincerely thank you for your careful reading and revision of the manuscript. Regarding the comments and suggestions proposed in the review, we made the following changes.

Comment: (The manuscript entitled “Investigation on the seismic wave propagation characteristics excited by explosion source in high-steep rock slope site using discrete element method” presents an interesting study that may provide scientific knowledge regarding the dynamic response of rock slope sites to seismic waves produced by explosive material for rock breaking during tunnel construction. The study specifically focuses on determining the way different points of a granite site react to seismic waves produces by a blast close to the slope. The manuscript is well written, although a review of the English language is recommended, since several grammar and syntax issues were detected. The structure of the text is well organized, although small reconfiguration is recommended. Figures and tables are well displayed and sufficiently described to support the conducted survey, with few minor exceptions that have been pointed in the specific comments to the authors. The bibliography is up to date and the appropriate citations exceed where needed, with few minor exceptions that have been pointed in the specific comments to the authors. There are several corrections and clarifications that have to be made prior to publication in the Sustainability Journal, therefore Minor Revision is suggested.)  

  1. Comment:(General comment: Please separate the chapters by adding main Chapter at the beginning. For example for Chapter “3. Wave propagation characteristics of seismic waves triggered by explosion source in the time domain” add first Chapter 3 Results and after the rest “3. Wave propagation characteristics of seismic waves triggered by explosion source in the time domain”.  Please apply the same for Chapter 4 Discussion, for Chapter 5 Conclusions …  and so on.

Response: We have made correction according to the Reviewer’s comments.

Abstract

  1. Comment:(Line 16 : “ … construction ”. Please add a sentence after “construction” stating what the goal of the study is.)

Response: We have made corrections according to the Reviewer’s comments. 

“To investigate the wave propagation characteristics and attenuation law of seismic waves induced by explosive sources in rock sites from the perspective of time and frequency domains, the high performance matrix discrete element method (MatDEM) is used to carry out numerical simulation tests on a granite rock medium site.”

  1. Comment:(Line 25 : “The … to Line 29 … region.” Please break the long sentences into shorter ones that have one meaning. Please apply the same for the rest of the manuscript.)

Response: We have made corrections according to the Reviewer’s comments.

“The slope region can weaken the dynamic response of seismic waves generated by the explosion source. In particular, the high-frequency band (>20 Hz) has an obvious filtering effect. The dynamic response of the P-wave induced by the explosive source is greater than that of the S-wave in the bedrock and surface region.”

Introduction

  1. Comment:(General comment: The Introduction is a bit too long. Please make it shorter. In addition, a review of the English language is recommended, since several grammar and syntax issues were detected.)

Response: We have made corrections according to the Reviewer’s comments. We have simplified the Introduction. We have checked the English language. Please check it.

  1. Comment:(Line 55: “ Many scholars … to line 73 … and so on”. Please do not include a list of what each scholar has done. Either remove the entire paragraph or try to combine the work that have done and how does that connect or add to your study. This segment should be included in the Discussion Chapter.)

Response: We have made corrections according to the Reviewer’s comments.

“At present, field blasting experiments, vibration tests, and numerical simulations are mainly used to analyze and investigate the propagation characteristics and attenuation law of blasting vibration [8]. In particular, numerical simulation has become one of the commonly used methods in the field of explosion impact [8-9]. The finite element method (FEM), finite difference method (FDM), and discrete element method (DEM) are commonly used in numerical simulation of explosion impact [10-11]. Some scholars have researched the propagation characteristics and attenuation law of blasting seismic waves by using FEM numerical simulation method [12-13]. FEM has great limitations for discontinuous media, infinite domain, large deformation, and stress concentration. Aiming at the large deformation of discontinuous media, some scholars began to use the FDM to investigate the dynamic response characteristics of rock-soil mass [14-15]. However, the FDM is difficult to simulate the failure process of the rock-soil mass because of its arbitrary division and boundary conditions. Some scholars began to use the DEM to study the dynamic response law of rock slopes, such as the PFC2D [16], UDEC [17], and so on. Moreover, MatDEM has been applied to geotechnical engineering by some scholars, in particular, the slope disaster process analysis has been discussed via MatDEM. Chen and Song used MatDEM to study the evolution process and failure mode of loess bedrock landslide [18]. Chen et al. used MatDEM to simulate the failure evolution process of reservoir bank landslide and forecast the danger areas affected by reservoir bank landslide [19]. Li et al. Used MatDEM to study the deformation and failure process and evolution of fractured rock mass [20]. The results show that the DEM is mainly suitable for solving discontinuous media and large deformation problems, and can better simulate the dynamic response and failure process of complex rock-soil mass. Therefore, many achievements have been made in the numerical simulation of explosive source seismic waves in previous studies. However, due to the limitations of the previous DEM, such as particle number, computing speed, and computing efficiency, a more efficient and convenient discrete element numerical simulation method is urgently needed.”

  1. Comment:(Line 96 Wu et al. … to Line 102 … geological conditions. Same comment as for comment (4))

Response: We have made corrections according to the Reviewer’s comments. Please check it.

“Some scholars have researched the influence of landforms on the propagation law of blasting vibration waves, and the research results mainly focus on the blasting vibration and attenuation law under flat terrain, the attenuation effect of concave landforms, the amplification effect of convex landforms, etc [25-27].”

  1. Comment:(Line 111: For Figure 1 there is citation [33] however in the context Line 86 the citation is [25]. Please add the right citation in both places.)

Response: We have made corrections according to the Reviewer’s comments. Please check it.

  1. Comment:(Line 128 : Please add after “…application value” 2 sentences, one stating what is the main goal of the study and one stating what are the main objectives.)

Response: We have made correction according to the Reviewer’s comments.

“This work takes the typical granite lithologic medium site as the research object. The research mainly focuses on the propagation characteristics of seismic source wavelets in the rock slope site under the blasting effect of explosives. ”

  1. Comment:(General comment: Please make a more clear definition and elaboration of the methods for this study by separating them in distinct chapters. For example, use a chapter to present the “Fast GPU Matrix computing of the Discrete Element Method (MatDEM)” another chapter to present “The discrete element numerical model of high-steep rock slope” …  “Discrete element dynamic analysis of the rock medium” “The time-frequency analysis method” and so on …

Wave propagation characteristics of seismic waves triggered by explosion source in the time domain Propagation characteristics of the seismic wave field)

Response: We have made corrections according to the Reviewer’s comments. Please Check them.

  1. Comment:(Line 297: “Figure 6”. Figure 6 is the same as figure 5. Please remove figures 6 and add the right ones, if applicable.

Waveform characteristics of waves triggered by explosion source)

Response: We have made corrections according to the Reviewer’s comments. 

We are very sorry that we put the picture in the wrong place due to our negligence. We have corrected it. Please check it.

(a) t=0.038 s                                     (b) t=0.076 s

(c) t=0.114 s                                     (d) t=0.152 s

Figure 6. Distribution characteristics of wavelet displacement field of explosion source in the DEM model.

  1. Comment:(Line 303 : Figure 7 must be placed after Chapter 3.2 Waveform characteristics of waves triggered by explosion source. In addition, consider presenting the figures in this Chapter based on the different points MP-27 only … and so on, rather showing them together as it looks cluttered and it is difficult to follow the results presented in the text in the figures. If decided to change the formatting of the figures please apply for the entire manuscript.Dynamic response characteristics of S- and P-waves of seismic wave triggered by explosion source )

Response: We have made corrections according to the Reviewer’s comments. Please check them.

(a)                                              (b)

(c)                                             (d)

Figure 15. Wavelet peak value distribution of explosion source: (a) PGVx; (b) PGVz; (c) PGAx; (d) PGAz.

(a)                                            (b)

Figure 20. Distribution of peak Fourier spectrum amplitude (PFSA) of acceleration wavelet excitation from the explosion source: (a) PFSAx; (b) PFSAz.

  1. Comment:(Line 384: “It can … to Line 387 … P-wave”. Please move this section to Discussion as this part is supposed to be Results where there is only presentation of the results without explanation.)

Response: We have made corrections according to the Reviewer’s comments. Please check it.

  1. Comment:(General comment : Please try to put the Figures used in the manuscript closer to the text where they are mentioned. For example Line 419 mentions “Fig.15.” which can be found almost after 2 pages. Please try to bring it closer. Same with the rest of the Figures used in the manuscript.)

Response: We have made corrections according to the Reviewer’s comments. Please check them.

  1. Comment:(General comment : Most of Chapters 4 and 5 are results and should be included with Chapter 3 in one single Chapter, Results. In addition, there is no Discussion Chapter which is necessary in order for a manuscript to be published. Discussion is probably the most important Chapter from a publication as it elaborates on the findings, how do they connect to previous studies and what is the major contribution of this study to the Engineering and Construction Sectors.  For example read comment 5, in addition Line 480 “ This indicates … to line 484 … as an example.” should be part of Discussion. Also Line 384 “It can … to line 386 … waves” this is also Discussion.)

Response: We have made corrections according to the Reviewer’s comments.

We have rewritten the conclusion section. Please check them.

4. Discussion

In this work, the seismic wave propagation characteristics of granite slope sites under blasting are studied by MatDEM. From the perspective of time and frequency domains, the dynamic response characteristics and attenuation law of seismic waves in slope and bedrock areas are discussed from the perspective of multi-parameter and multi-angle. The result of time-domain analysis mainly shows the propagation characteristics of seismic waves under blasting from the perspective of mechanical deformation. The frequency-domain analysis mainly reveals the propagation characteristics of seismic waves from the perspective of inherent characteristics. The analysis results in the time and frequency domains are complementary to each other. In addition, based on MatDEM numerical simulation, the dynamic response characteristics of blasting-induced seismic waves can be systematically analyzed from the perspective of the velocity field, displacement field and energy transmission. This work can provide an important reference for the dynamic response of blasting-induced seismic waves in rocky sites. However, in this work, MatDEM was used to study the propagation characteristics of explosion-induced seismic waves in a granite site, which is a special case. For other rocky sites, the wave propagation characteristics of explosion-induced seismic waves need to be discussed, and further numerical simulation of sites with different lithologies is also needed. At the same time, this study also needs to use field test and laboratory model tests for further verification, which will be the focus of the next research.”

  1. Comment:(General comment : Most of the things included in Chapter 6 Conclusions are not actual conclusions but more so Results. The conclusions come out of the Discussion Chapter. Since there is no Discussion Chapter there cannot be any Conclusions. So please write a Discussion Chapter first and after that write the Conclusions Chapter.)

Response: We have made corrections according to the Reviewer’s comments.

We added the discussion section and rewrote the conclusion section. Please check them.

5. Discussion

In this work, the seismic wave propagation characteristics of granite slope site under blasting are studied by MatDEM. From the perspective of time and frequency domains, the dynamic response characteristics and attenuation law of seismic wave in slope and bedrock area are discussed from the perspective of multi-parameter and multi-angle. The result of time-domain analysis mainly shows the propagation characteristics of seismic waves under blasting from the perspective of mechanical deformation. The frequency-domain analysis mainly reveals the propagation characteristics of seismic waves from the perspective of inherent characteristics. The analysis results in the time and frequency domains are complementary to each other. In addition, based on MatDEM numerical simulation, the dynamic response characteristics of blasting-induced seismic waves can be systematically analyzed from the perspective of the velocity field, displacement field and energy transmission. This work can provide an important reference for the dynamic response of blasting-induced seismic waves in rocky sites. However, in this work, MatDEM was used to study the propagation characteristics of explosion-induced seismic waves in a granite site, which is a special case. For other rocky sites, the wave propagation characteristics of explosion-induced seismic waves need to be discussed, and further numerical simulation of sites with different lithologies is also needed. At the same time, this study also needs to use field tests and laboratory model tests for further verification, which will be the focus of the next research.

  1. 6. Conclusions

MatDEM is used to study the wave propagation characteristics and attenuation law of seismic waves triggered by explosion sources in the rock site. Some main conclusions can be drawn as follows:

  1. The analysis of the velocity and displacement fields of waves shows that the wave propagation in the rock slope site is characterized by the arc. With the explosion duration and the increase of the distance from the explosion source, the amplitude of waves attenuated gradually. The wave attenuation effect of the ground surface is smaller than that in the bedrock. Near the explosion source (<25 m), the PGA of waves on the right side of the explosion source is the largest but decays rapidly, while the PGA on the left and lower side of the explosion source is smaller, in particular, the PGA on the lower side is the smallest.
  2. Time and frequency domain analysis shows that the waves attenuate gradually with the increase of the distance from the explosion source. The explosion source has the greatest influence on the dynamic response within the range of 25 m, and the influence is small when it is beyond 100 m from the explosion source. The dynamic response of waves is larger in the range of 25 m from the explosion source, and the attenuation rate is the fastest. The attenuation rate is small and tends to be stable in the area of 100 m away from the explosion source. The slope region can weaken the dynamic response of waves, especially filtering the seismic waves of the high-frequency band (>20 Hz), which greatly reduces the amplitude of the high-frequency spectrum. The seismic wave energy has a rapid attenuation effect within a short time (0-0.3 s), and the attenuation effect gradually decreases and becomes stable when the blasting time is >0.3 s.
  3. The existence of a slope changes the dynamic response characteristics of waves. In the bedrock and the ground surface, the dynamic response of the P-wave is greater than that of the S-wave, and the PGV and PGA of the P-wave are greater than that of the S-wave. The PGV and PGA of the S-wave in the slope region are smaller than those of the P-wave, and the dynamic amplification effect of the P-wave is larger than that of the S-wave. The waves in the slope region show an attenuation effect along the slope surface. The PGV and PGA of waves in the slope interior increase along the increase of elevation, which has a typical elevation amplification effect.”

Special thanks to you for your helpful comments.

We tried our best to improve the manuscript and made some changes to the manuscript. These changes do not influence the content or framework of the paper. We did not list here all of the changes, but we marked them in the revised manuscript. We appreciate the work of the Editor and Reviewers and hope that the corrections will be met with your approval. Once again, thank you very much for your comments and suggestions.

Yours Sincerely,

Prof. Menxin Liu

School of Civil Engineering, Northeast Forestry University, Harbin 150040, China

Corresponding author. E-mail address: liumengxin@nefu.edu.cn (Mengxin Liu)

Prof. Chun Liu

School of Earth Sciences and Engineering, Nanjing University, Nanjing 210023, China

Corresponding author. E-mail address: chunliu@nju.edu.cn (Chun Liu)
